# AmadeusGPT: a natural language interface for interactive animal behavioral analysis

**Shaokai Ye**
EPFL
Geneva, CH

**Jessy Lauer**
EPFL
Geneva, CH

**Mu Zhou**
EPFL
Geneva, CH

**Alexander Mathis**
EPFL
Geneva, CH

**Mackenzie W. Mathis**
EPFL
Geneva, CH
mackenzie.mathis@epfl.ch

## Abstract

The process of quantifying and analyzing animal behavior involves translating the naturally occurring descriptive language of their actions into machine-readable code. Yet, codifying behavior analysis is often challenging without deep understanding of animal behavior and technical machine learning knowledge. To limit this gap, we introduce AmadeusGPT: a natural language interface that turns natural language descriptions of behaviors into machine-executable code. Large-language models (LLMs) such as GPT3.5 and GPT4 allow for interactive language-based queries that are potentially well suited for making interactive behavior analysis. However, the comprehension capability of these LLMs is limited by the context window size, which prevents it from remembering distant conversations. To overcome the context window limitation, we implement a novel dual-memory mechanism to allow communication between short-term and long-term memory using symbols as context pointers for retrieval and saving. Concretely, users directly use language-based definitions of behavior and our augmented GPT develops code based on the core AmadeusGPT API, which contains machine learning, computer vision, spatio-temporal reasoning, and visualization modules. Users then can interactively refine results, and seamlessly add new behavioral modules as needed. We used the MABe 2022 behavior challenge tasks to benchmark AmadeusGPT and show excellent performance. Note, an end-user would not need to write any code to achieve this. Thus, collectively AmadeusGPT presents a novel way to merge deep biological knowledge, large-language models, and core computer vision modules into a more naturally intelligent system. Code and demos can be found at: `https://github.com/AdaptiveMotorControlLab/AmadeusGPT`.

## 1 Introduction

Efficiently describing and analyzing animal behavior offers valuable insights into their motivations and underlying neural circuits, making it a critical aspect of modern ethology, neuroscience, medicine, and technology [1, 2, 3, 4]. Yet behavior is complex, often multi-faceted, and context-dependent, making it challenging to quantify and analyze [5, 6]. The process of translating animal behavior into machine-readable code often involves handcrafted features, unsupervised pre-processing, or neural network training, which may not be intuitive to develop for life scientists.

To understand animal behavior one needs to complete a series of sub-tasks, such as obtaining animal poses and identities, object locations and segmentation masks, and then specifying events or actions the animal performs. Significant progress has been made in automating sub-tasks of behavioral analysis such as animal tracking [7, 8, 9], object segmentation [10, 11], and behavior classification [3, 12, 13, 14, 4], yet behavioral phenotyping requires additional analysis and reasoning [2, 15, 16, 13,

37th Conference on Neural Information Processing Systems (NeurIPS 2023).

17, 8]. This is typically done with feature computations such as measuring time spent in regions of interest or general statistics of locomotion [13, 8, 18].

The challenge of making behavior analysis accessible and interpretable is hindered by the difficulty of combining task-specific models and the lack of an intuitive natural language interface to produce machine code. In an ideal scenario, a behavior analysis practitioner would be able to explore behavioral data, define their desired actions using natural language, and visualize captured behaviors without needing to learn how to train models, write scripts, or integrate code bases. Our framework, AmadeusGPT, takes the first step towards achieving this goal. AmadeusGPT provides a natural language interface that bridges users' prompts and behavioral modules designed for sub-tasks of behavior analysis. Our results show that AmadeusGPT outperforms machine learning-based behavior analysis classification tasks in the MABe [4, 19] benchmark by using prompts highly similar to their official definitions of each behavior, namely with small modifications and only three tunable parameters.

AmadeusGPT offers a novel human-computer interaction approach to behavior analysis, providing a unique user experience for those interested in exploring their behavioral data. Through natural language prompts, users can ask questions, define behaviors on-the-fly, and visualize the resulting analyses plus the language output (Figure 1). We show that with our novel dual-memory mechanism, defined behaviors are not lost (when exceeding the token limit), wording can be automatically rephrased for robustness, and the state of the application can be restored when relaunched, providing seamless and intuitive use (Figures 1-4).

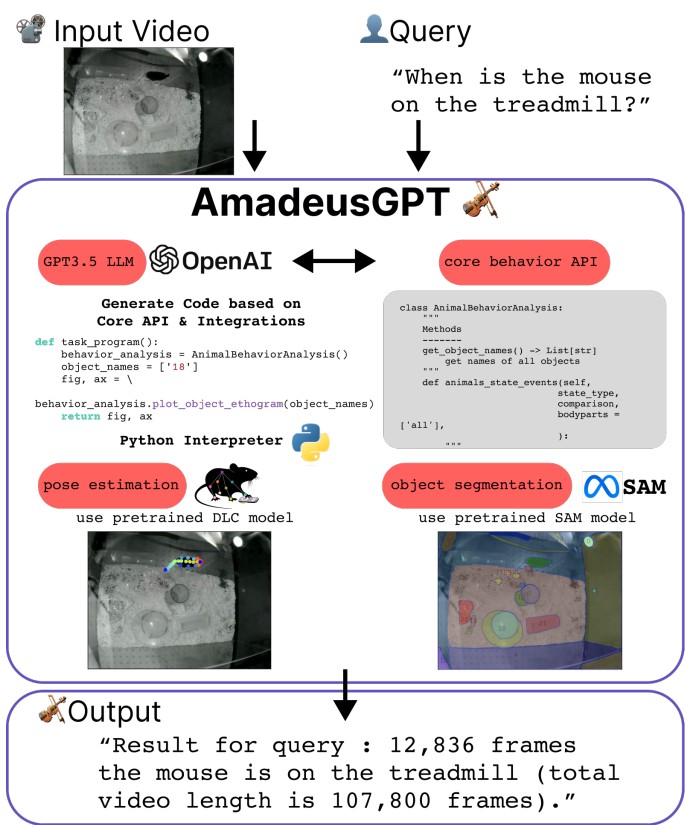

Figure 1: **AmadeusGPT**: a LLM, computer vision, and task reasoning platform for animal behavior. Users input video and prompts and AmadeusGPT queries our API docs and reasons about which models and analysis to deploy.

To capture animal-environment states, AmadeusGPT leverages state-of-the-art pretrained models, such as SuperAnimals [17] for animal pose estimation and Segment-Anything (SAM) for object segmentation [11]. The platform enables spatio-temporal reasoning to parse the outputs of computer vision models into quantitative behavior analysis. Additionally, AmadeusGPT simplifies the integration of arbitrary behavioral modules, making it easier to combine tools for task-specific models and interface with machine code.

## 2 Related Work

**Large Language Models.** In recent years there has been a surge in the development of large language models (LLMs) [20, 21, 22, 23] that show exceptional abilities in natural language processing tasks such as language generation, interaction, and reasoning. One notable example is ChatGPT, which is built on GPT-3+ and trained on a massive corpus of text data [22]. ChatGPT has demonstrated impressive performance in a wide range of natural language processing tasks, including text classification, language modeling, and question-answering. In addition to language processing, LLMs have

also been applied to other domains such as image and video processing [24], audio processing, and natural language generation for multi-modal inputs [24, 25].

**LLM Integrations.** Recent works leverage LLMs to generate code for computers to execute, including visual programming with VisProg [26] and ViperGPT [27], and HuggingGPT [28] that exploits pre-trained AI models and smartly handle API-GPT interactions [26, 27]. However, animal behavior analysis lacks a natural language-computer vision pipeline and requires diverse open-source code-base to complete the sub-tasks. Concurrent work by Park et al. [29] on Generative Agents uses a dual-memory system and iterative chatting, but has a more expensive execution flow compared to our approach, which uses a memory mechanism with symbols as pointers.

**Behavioral Analysis.** Machine learning and computer vision techniques have increasingly been employed in animal behavior analysis in recent years [7, 9, 12, 30, 31, 32]. These techniques offer advantages such as automation, high throughput, and objectivity, compared to traditional methods that rely on manual observation and annotation [1]. Deep learning models, such as convolutional neural networks (CNNs) and transformers, have been utilized for feature extraction and classification of animal behaviors in various domains such as social behavior, locomotion, and posture analysis (reviewed in [2]). Yet, universal interfaces to the plethora of pose estimation tools and downstream analysis methods are lacking.

We used a rule-based approach that leveraged the pose estimation outputs to compute behaviors, similar to the approach used in LiveMouseTracker (LMT) [33]. However, unlike LMT, which requires users to interact with code and requires a specific hardware to collect data, AmadeusGPT is agnostic to data collection and provides a natural language interface that is easy to use and customize. Additionally, unlike LMT, AmadeusGPT includes object segmentation, enabling it to capture animal-object interactions.

**Task Programming for Behavior.** Task programming for behavior analysis [30] aims to leverage the domain knowledge from experts to help extract the most relevant features that are useful for the downstream behavior classification task. However, task programs were only considered as Python functions that help extract better features (i.e., features built from poses). In this work, we generalize the scope of task programming: any sub-task that can be achieved as machine-executable Python code is considered task programs in AmadeusGPT, and can be queried and developed with natural language. Thus, we name the generated function in AmadeusGPT (target) task programs.

# 3   AmadeusGPT

Figure 2: **Schematic of AmadeusGPT design and features**.

AmadeusGPT is an interactive human-computer platform that allows users to describe in natural language the behavioral analysis they want to be performed. It utilizes ChatGPT as the user-guided controller and a range of machine learning and computer vision models as collaborative executors to analyze animal behavior, starting from raw video and leveraging new pretrained pose estimation models that can run inference across species and settings [17], and powerful objection segmentation models (SAM [11]) (Figure 2). Here we focus primarily on mice, but also show it works for horses to demonstrate that nothing precludes AmadeusGPT to be used on a range of animals.

AmadeusGPT uses LLMs to generate Python executable code that fulfills user-specified queries in the prompt. Building such a system requires LLMs to learn to manipulate core process resources in a constrained way. If the user's prompt is unclear or beyond the system's capacity, the generated code might result in errors that require programming expertise. Intuitive error messages are therefore essential for ensuring a consistent natural language experience, while maintaining the ability for users

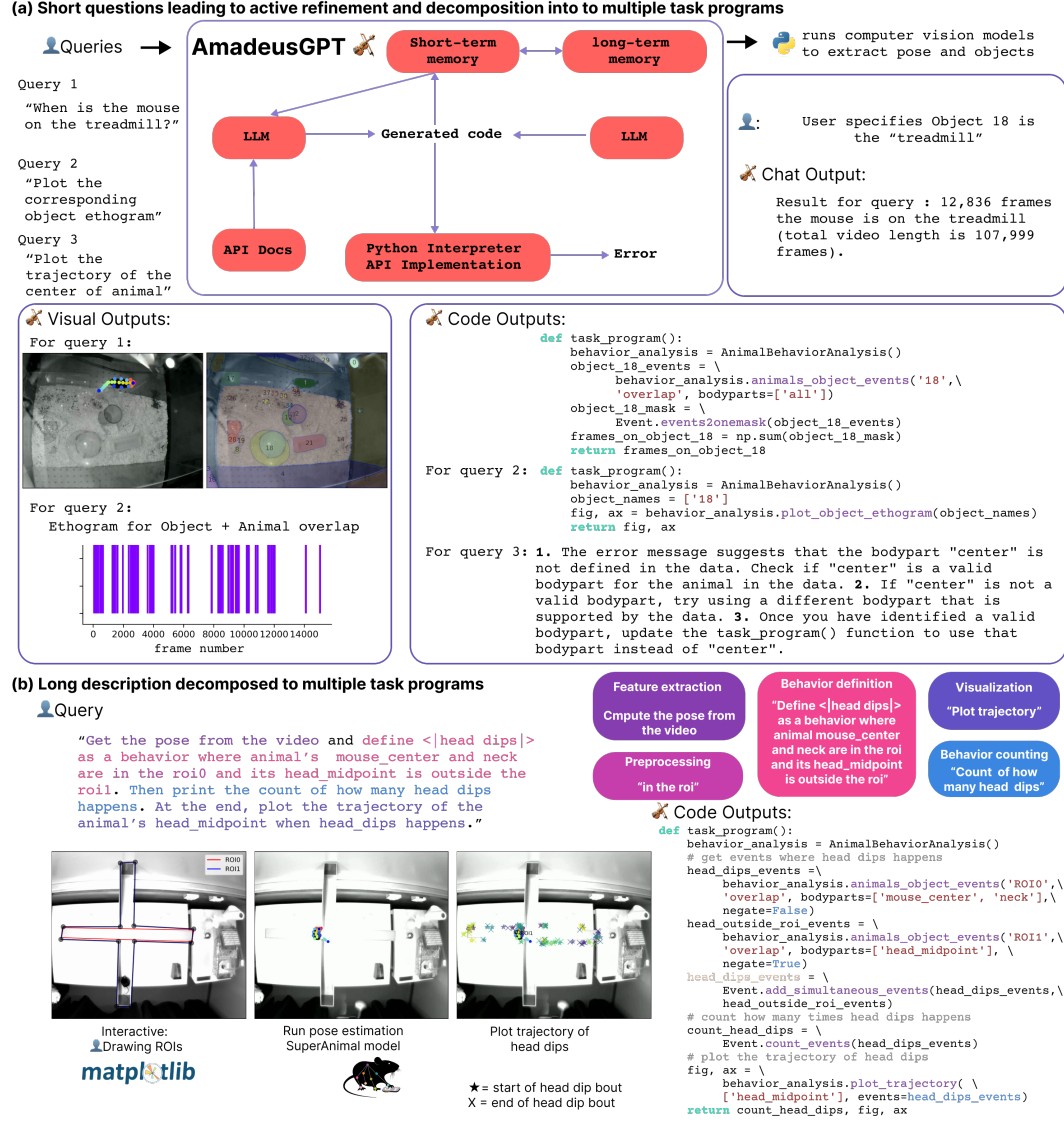

Figure 3: **AmadeusGPT: a natural language model enhanced behavioral analysis system. (a)** Users can start by uploading a video and asking a question to AmadeusGPT about what they want to do. AmadeusGPT will run computer vision models if the target task depends on pose estimation (e.g., the posture or location of the animal) and object segmentation (e.g., when does the animal overlaps with an object) and/or ask clarifying questions. Once correct, the user can also save these queries/code outputs. Three example queries and outputs are shown. **(b)** Alternatively, users can provide a detailed recipe for how they want the data analyzed. The colored text highlights action-items for AmadeusGPT (matching the colored boxes).

to iteratively refine generated code with language. Therefore, to build AmadeusGPT we leveraged GPT3.5, augmented it, developed a dual-memory system and core behavioral utilities that together make AmadeusGPT an end-to-end, human language-AI system (Figure 2).

## 3.1 LLMs as natural interfaces for code development and execution

ChatGPT (i.e., with GPT3.5 or 4) [23] is able to generate code that corresponds to users' prompts due to instruction tuning paired with access to a large amount of code in its training data. However, the native ChatGPT/GPT3.5 or 4 API cannot serve as the language interface for behavior analysis for the following reasons: (1) it cannot work with private APIs that are not in its training data; (2) it

hallucinates functions if too many implementation details are exposed or asked to be provided; (3) the context window does not have a sufficient capacity size for complex tasks that require reading large source code files; (4) it has no Python interpreter to execute the code it suggests. Therefore, we built an augmented version of GPT to overcome these limitations (see Sections 3.2 and 3.3).

## 3.2 Augmented-GPT and API design

AmadeusGPT sends its API documentation to ChatGPT in order to smartly constrain the output code and immediate Python execution and thereby always augments the data available to GPT3.5 or GPT4. In the API documentation, implementation details are encapsulated and only the function documentation is exposed (see Appendix for example API docs). Importantly, there is an "explanation prompt" with each function example that serves as a hint for GPT3.5 to understand what the API is for. This separation of API documentation and implementation has two advantages: it reduces token use, and prevents hallucinating resources that do not exist in the core functions or modules linked to AmadeusGPT. Concretely, we prompt GPT3.5 to strictly follow our API (see Appendix). This design improves the reliability of code generation while imposing quality control.

We followed three core principles when designing the API. Firstly, we developed a code with an *atomic API* that consists of simple functions designed to perform specific operations (Figure 2). These functions are modular, reusable, and composable, making them easy to combine with other API functions to create more complex functionality. The use of atomic API improves the readability of the code for users and developers and reduces the chance of LLMs confusing ambiguous API calls. Secondly, we leverage polymorphism in code design to generalize our API to variants of a sub-task routed by parameter values and input types since it is not desirable nor realistic to provide an example code for every possible sub-task. Thirdly, to make AmadeusGPT cover a range of behavioral analysis tasks, we identify *core behavioral modules* that cover common behavior analysis sub-tasks and additionally use integration behavioral modules for task-specific sub-tasks.

## 3.3 Dual Memory Mechanism

As GPT3.5 is implemented with a transformer architecture that has a context window size limitation of 4,096 tokens for hosting the history of conversations [22]. This would limit how far back AmadeusGPT can use the context, such as a user-defined behavior or its own generated code. We demonstrate in Figure 4 that without tackling the context window limitation, forgetting will happen, which results in wrong code that hallucinates functions or variables that do not exist.

To tackle this limitation, we implemented a dual-memory mechanism that consists of both short-term memory and long-term memory banks. In this work, short-term memory is implemented as a dynamic deque that holds the chat history, pulling in new tokens and removing older ones when full (i.e., beyond the 4,096 tokens). At inference time, all text in the deque is sent to the ChatGPT API call.

Long-term memory is implemented as a dictionary in RAM. The keys of such a dictionary are called symbols in this work. We define read and write symbols to instruct the communication between short-term memory and long-term memory. With the help of regular expression, a symbol that is enclosed by $< || >$ triggers a memory writing operation that writes the context of the corresponding chat into the dictionary using the symbol name as the key. As in Figure 3, keyword $< | \text{head dips} | >$ triggers memory writing that stores the whole sentence under the key name "head dips". Similarly, $<>$ triggers a reading operation that pushes the stored sentence into the front of short-term memory with an additional prompt to remind GPT3.5 that the retrieved memory is used for context instead of a formal instruction. This allows a read operation retrieval from the long-term memory to be used as a context for a new instruction. Additionally, with the long-term memory, AmadeusGPT can store its states into disk and the user can restore the states after re-launching the application (See Appendix). Collectively, we call this augmented-GPT3.5.

## 3.4 Core Behavioral Modules

The language query feature of ChatGPT inspired us to provide a more natural way for humans to perform behavior analysis. Namely, it provides users with a platform to perform behavior analysis by asking questions or instructing tasks in an interactive manner. We imagine that users will either ask a question or provide longer instructions to complete a task (Figure 3). In addition, if users ask follow-up questions to a previous answer from AmadeusGPT, it attempts to answer with executable code

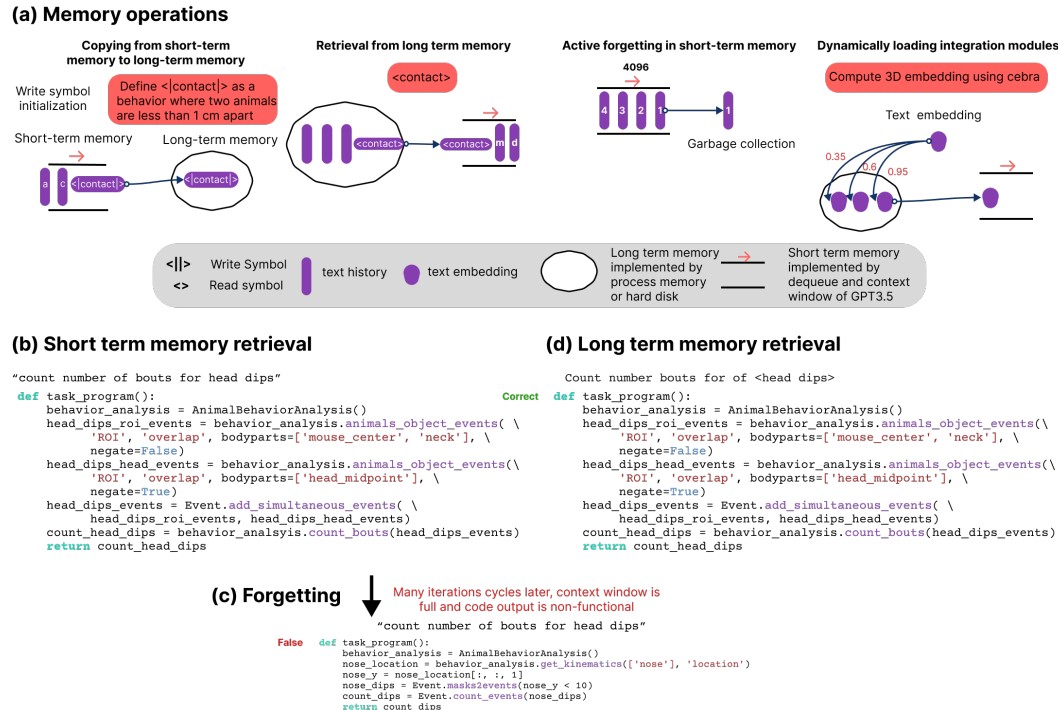

Figure 4: **Dual Memory Mechanism for augmenting GPT3.5 or 4. (a)** We introduce a long-term memory module that overcomes running out of tokens, and a dynamic loading system for code integrations such as for advanced uses like dimensionality reduction with UMAP [34] or CEBRA [35] (see Appendix). **(b)** An example query and output from short-term memory only (early in an interactive session), **(c)** if long-term memory is ablated after running out of tokens the output is non-functional. **(d)** Yet, with our new memory system, it can easily retrieve the identical, correct output within a long session or upon restarting.

called "task programs" that are executed by the backend Python interpreter. In the API documentation we mostly specify code examples for uni-purpose task programs (such as counting events). However, we show in Figure 3 that our augmented-GPT3.5 API is able to compose multi-purpose task programs (such as computing events and interactions with objects over time to produce plots).

Figure 3b shows how the user can give a complex instruction that covers multiple sub-tasks, including pose extraction, behavioral definitions, interactively drawing regions of interest (ROIs), then visualizing and performing tasks, such as behavior event counting. In this case, AmadeusGPT is able to decompose the description into multiple task programs and assemble the final program. Alternatively, the user can also ask a question "When is the mouse on the treadmill?" or "Count the number of head dips per ROI" as follow-up queries to AmadeusGPT's answer, or change the color map of the previous plot, etc. This all relies on the core behavioral modules and their ability to be combinatorically used (Figure 2).

Our core behavioral modules try to cover the most common behavioral sub-tasks. Generally speaking, a typical behavior analysis task asks question about what animals do through a spatio-temporal space. This can simply be the amount of time that an animal spends moving in a particular ROI, to more advanced analysis like measuring animal-animal or animal-object interactions. This requires varying levels of key-point tracking, object segmentation, and video (temporal) reasoning. Thus, we implemented the following core modules and always send them with the queries during augmented-GPT API calls (Figure 2).

- *Kinematic feature analysis.* As many behaviors are concerned with the kinematics of bodyparts, we provide a module for kinematics analysis. This includes filtering the pose data, calculating speed, velocity, and acceleration. It also includes movement-based statistical analysis, such as queries related to computing the amount of time spent locomoting (i.e., a velocity over some minimal threshold computed across the video, or time spent in an ROI).

*- Animal-object environment states.* To capture animals' and environment states, we deploy pretrained computer vision models such as pose estimation with SuperAnimal models [17, 7] and objects with SAM [11]. Note they can be easily substituted by tailored supervised models. To support end-to-end behavior analysis with raw video only, we use them as the default models for their wide coverage of animals and objects. We also implemented code infrastructure that abstracts "static objects" and "moving objects" to represent objects in the environment as well as customized ROI objects and animals respectively. The animal-object relation and animal-animal relation are modeled and saved in lookup tables. These relations mostly cover binary spatial relations such as "to the left", "to the right", "overlap", "orientation", and numerical spatial relations such as "distance", "angle" (i.e., the angle between animals, based on computed neck-to-tailbase body axes) and "gazing angle" (i.e., the angle between animals' head coordinate systems, determined from the nose-neck line) (see Appendix).

*- Spatio-temporal reasoning.* After computer vision models track animals and segment objects, their outputs are used to build internal spatio-temporal relation tables among animals-animals and animals-objects. We provide code infrastructures to support queries of events (i.e., sequential spatio-temporal relations and simultaneous spatio-temporal relations). Users can query events where animals move from one state to the other (see also Figure 5b).

### 3.5 Integrations Beyond Core Behavioral Modules

Integration modules aim to cover task-specific behavior analysis sub-tasks such as dataset loading, embedding calculations, and visualization tools. Because they are task-specific, we do not send the API documentation of these modules. Instead, we rely on "dynamic loading" to load only few of the selected API documents for integration modules at inference time. To allow for dynamic loading, we use the embedding API from OpenAI to turn API documents of integration modules into text embedding vectors and cache them in RAM or disk. The user's prompt then acts as a query to retrieve $k$ (a hyperparameter) most relevant integration modules by cosine similarity (Figure 4). We also allow users to manually load integration modules by prompting "loading module \module-path" if users want to save the cost of using the embedding API and/or they are creating their own modules.

**Error handling.** We include error handling to help users understand when instructions are beyond the system's capabilities or ambiguous. Here, AmadeusGPT forwards prompts, error messages, and API docs to the ChatGPT API for natural language explanations (Figure 3a, Query 3 example).

**Rephraser.** Users can ask questions with language that might be very different from those in our API docs, which can cause performance degradation due to out-of-distribution prompts (See Section 4). To overcome this we leverage a native GPT3.5 (i.e., without our augmentation) that we call "Rephraser" that is tasked to turn users' expressions into a style that is similar to that found in our API docs. We wrote in the system prompt of Rephraser a few examples of such rephrasing, hoping the GPT3.5 can learn it via few-shot learning. Therefore, Rephraser is tasked to act as test-time domain adaptation component for our system.

**Self-correction.** There are incidences where ChatGPT API makes obvious mistakes that cause runtime errors, therefore we implemented a self-correction mechanism to help ameliorate this. When there is an error executing the generated python code, we forward the error message and the output of our error handler to an independent ChatGPT API connection with a system prompt that encourages it to revise the function. The number of times for such retrying is flexible, but setting this to three in practice generally improves the success rates (See Appendix for an example). We keep the history of retries in the context window to help it to learn from failures if it takes more than one trial to correct the code. Note that ChatGPT API does not have read/write access to the implementation of our APIs thus self-correction cannot correctly revise the code if the errors stem from our code.

**Explainer module.** AmadeusGPT takes users' queries and generates Python code to address the queries. The executed function returns results of multiple data types, including plots, strings, numpy arrays, etc. However, it might not always be straightforward for users to link those returned results to the queries. Moreover, the users do not know how much they can trust the returned results. In many cases, checking the generated code can help. However, inspecting the code requires python programming knowledge. Therefore we add and explainer module, which is another LLM whose job is to explain to the users how to link the results to the queries. The explainer is implemented as an independent ChatGPT API connection with its independent system prompt. In the system prompt, we ask the explainer to take the thought process parsed from the code generator, the return values

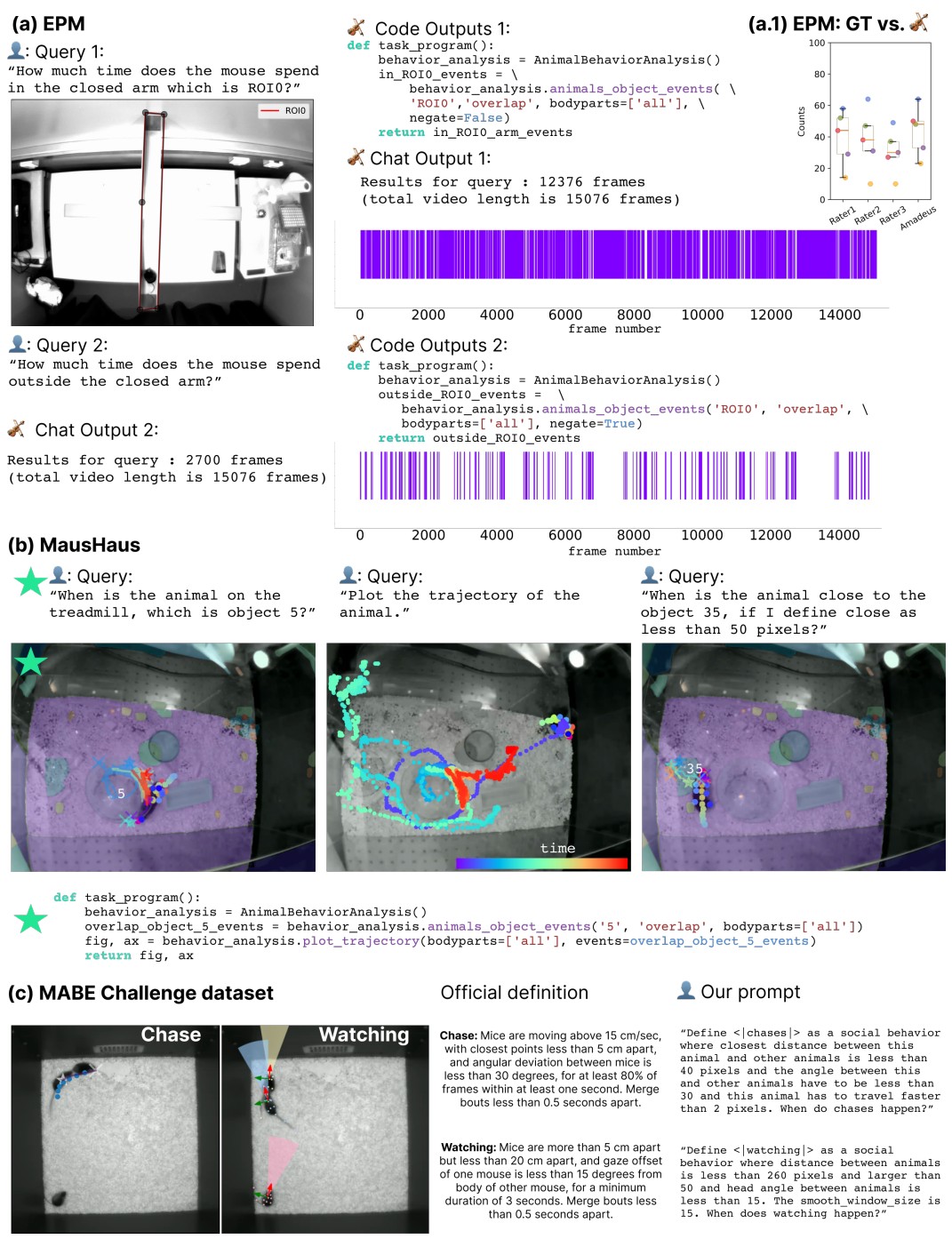

Figure 5: **Results on classical behavioral tasks with AmadeusGPT**. **(a)** Result on EPM showing limited time in the open arms by the mouse. The raster plot is an ethogram where there is a tick for every related event in time, i.e., the mouse is in the closed arm or in the open arm. (a.1) shows AmadeusGPT counts vs. three human raters [13] across five videos (colored dots). **(b)** Animal-object interactions can be computed in natural home cage settings. **(c)** Behavioral images and original description given in MABe, vs. our prompts that produce the quantitative results shown in Table 1. For Chase, we visualize the chasing animal's trajectory that overlaps with the predicted mask and the chasing animal's trajectory for the ground-truth mask. For Watching, we visualize the visual cone for each animal from each animal's head coordinate (axis by neck-nose vector and its normal vector).

| Tasks | T4 approach | T5 chase | T6 close | T7 contact | T8 huddles | T9 * | T10 * | T11 * | T12 watch |
|---|---|---|---|---|---|---|---|---|---|
| PCA baseline | 0 | 0 | 0.13 | 0.0008 | 0 | 0 | 0 | 0 | 0 |
| Top-entry 1 [4] | 0.020 | 0.010 | **0.708** | 0.558 | 0.310 | 0.0056 | 0.015 | 0.013 | 0.182 |
| Top-entry 2 [4] | **0.026** | 0.050 | 0.7072 | 0.561 | 0.214 | 0.0084 | 0.029 | 0.023 | 0.159 |
| Top-entry 3 [4] | 0.022 | 0.029 | 0.655 | 0.515 | 0.249 | 0.0062 | 0.015 | 0.014 | 0.162 |
| BAMS [37] | 0.02 | 0.023 | 0.664 | 0.533 | 0.302 | 0.0045 | 0.0165 | 0.014 | 0.191 |
| AmadeusGPT | 0.014 | **0.274** | 0.700 | **0.572** | **0.380** | **0.05** | **0.05** | **0.024** | **0.600** |

Table 1: F1 scores by AmadeusGPT on several tasks from the MABe Behavior Challenge 2022 [4]. We do not use representation learning (the aim of the benchmark), only rule-based task programming. *T9: oral-ear-contact; T10: oral-genital-contact; T11: oral-oral-contact.

from python as well as the user queries to explain whether the code and its return values meet the queries (See Appendix for an example).

# 4 Experiments

To evaluate AmadeusGPT we ran a series of qualitative and quantitative experiments. We used three datasets that highlight standard behavioral neuroscience settings. The first is an open-access Elevated Plus Maze (EPM) dataset from Sturman et al. [13]. The second is called MausHaus and is a video of a mouse for one hour (108K frames) in an enriched home-cage setting [17]. The third is video data of three mice interacting from the MABe 2022 Challenge [4]. We also include a fourth in-the-wild horse dataset [36].

**EPM.** The EPM is a classical task in neuroscience research to test an animal's anxiety about being in exposed "open arms" vs. "closed arms" [13]. We show that with minimal prompts and ROI interactive plotting a user can measure these events (Figure 5a, also see Figure 3). Here, we query AmadeusGPT to report both open and closed arms and note that the resulting raster plots (ethograms) do not identify the same frames, as one expects if it is correct (i.e., the mouse cannot be in both states at once). We also show that AmadeusGPT counts the number of events similar to those reported in ground truth annotations by three raters across five different videos (Figure 5a.1).

**MausHaus: enriched mouse-cage monitoring.** Studying more natural behavior is becoming important in research settings. Here, we demonstrate intuitive prompts that run pretrained models (SuperAnimal-TopViewMouse and SAM) then query spatio-temporal relationships between the model outputs (Figure 5b). As SAM only provides object labels, we use object number or interactive clicking on the object to ground the analysis (see also Figure 3).

**Horse gait analysis.** AmadeusGPT has no innate mouse preference. Therefore to show another application we performed equine gait analysis [36]. We leveraged horse videos with ground truth keypoint annotations on every frame and used SuperAnimal-Quadruped [17], and found comparable results within AmadeusGPT to human-level labeling and stride analysis (see Appendix Figure 8).

**MABe 2022 Benchmark.** The benchmark had two rounds for the three mouse dataset. We used the more popular first round (evaluation split) and therefore provided the pre-computed keypoints as inputs to AmadeusGPT. In Figure 5c, we show how AmadeusGPT captures two representative tasks from MABe benchmark, Chase and Watching. We use text that is close to the original definition to define the behaviors we want to capture. Note the units in our prompt are pixels for distance, radians for degree and pixel per frame for speed. We tested nine behaviors and report the F1 score as computed in the MABe evaluation (Table 1). Our approach is purely rule-based, thus no machine learning is needed and only three parameters are needed to be given or tuned: a smoothing window size for merging neighboring bouts, a minimal window size for dropping short events, and the pixel per centimeter. Note that tasks that are hard for machine learning models are also hard for our rule-based approach (Table 1, Tasks 9-11).

We do not intend to formally claim state-of-the-art on the benchmark, as the goal was to evaluate unsupervised representational learning models. Nevertheless, we show that in practice one can use a text definition of behavior to capture behaviors that are on par with, or better than, the competitive representation learning approaches.

**Robustness tests and stress-testing AmadeusGPT.** Users might query AmadeusGPT with expressions that are very different from the explanation text we provided in the API documentation. A potential pitfall is that AmadeusGPT overfits to our (the developers) expressions and biases. To test how robust AmadeusGPT is, we crafted five out-of-distribution (OOD) base questions for an EPM video. Then we asked a native GPT3.5 (part of our reflection module, see Figure 2) to generate five variants of each OOD question. We manually checked whether AmadeusGPT generated consistently correct results on those 25 generated questions, and it did 88% of the time with our Rephraser module vs. 32% without the Rephraser. The total number of tokens consumed using the ChatGPT-API was 4,322 and 5,002 with and without using Rephraser, respectively. Note that in both cases, the consumed number of tokens is larger than the maximal context window size of 4,096. This also shows that AmadeusGPT passes the stress-test we set for it in two key ways: (1) The short-term memory deque correctly maintains the constrained size without getting an error from the OpenAI API due to maximal token size error; (2) The diverse questions in the short-term memory do not result in mode collapse or severe performance degradation.

In additional testing with naive users we found that better LLMs boost performance. In brief, we sampled 30 naive user prompts at random out of 362 submitted via an App we developed, and found an 18% error rate with GPT3.5 yet with GPT4 [38] only 10%). See the Appendix for examples.

## 5 Discussion

AmadeusGPT is a novel system for interactive behavior analysis. By providing a user-friendly interface and leveraging the power of language models, AmadeusGPT enables researchers to more efficiently and effectively study animal behavior. We believe this system offers a significant contribution to the field of behavioral analysis. The natural language interface of AmadeusGPT empowers non-experts to conduct advanced behavioral analysis with cutting-edge computer vision (such as with SAM [11] and SuperAnimal [17]), but its reliance on LLMs raises potential ethical concerns such as bias amplification. The use of a custom-designed API module in AmadeusGPT helps limit potential biases in outputs, but future work should consider how integration modules can introduce biases [39]. It also allows for domain-specific human-AI interactions [40, 41].

Our proposed natural language interface with the augmented-GPT3.5 shows promise, but there are limitations that need to be addressed in future work. These include enhancing robustness by reducing bias in prompt writing, enabling more goal-driven code reasoning to make AmadeusGPT more collaborative, and improving generalization by extending the available APIs and preventing ChatGPT from writing incorrect code. We also only support the English language, but multi-lingual support with be explored in the future. Collectively, we believe AmadeusGPT promises to open new avenues for both leveraging the utility of, and developing on, cutting-edge computer vision and machine learning tools by using intuitive language.

It is also worth noting that AmadeusGPT leverages ChatGPT API calls, which means we do not have access to the weights of the underlying LLMs thus we cannot yet fine-tune or augment the LLMs. Therefore, our dual-memory leverages the process memory and disks to augment the language model, in contrast to works that augment LLMs by assuming the access to the LLMs [42, 43]. Similarly, without fine-tuning the underlying LLMs, we constrain the behaviors of the LLMs only via in-context learning for both the code generator and rephraser. While recent models such as GPT4 [38], CLAUDE [44] continue to make context window bigger and in-context learning more powerful, deploying them can be slower and more expensive and a strategy of combining a smaller model and a bigger model is worth exploring [45]. In the future, it would be interesting to compare our in-process-memory with in-context memory in terms of reliability and cost. It is also interesting to compare fine-tuned LLMs vs. LLMs that are constrained via in-context learning for future works that use natural language interface for behavior analysis.

Of course, AmadeusGPT is not limited to using GPT3.5 or GPT-4, and new models such as CLAUDE [44] or others could be used. However, the larger the model the slower the response time and the higher the computational cost. Thus, while these excellent works have started to overcome token limits, there are still limits such that our computationally efficient dual-memory system will be of broad interest to those developing both domain-specific solutions (such as AmadeusGPT) and generalist models (the base GPT models). Collectively, we believe AmadeusGPT promises to open new avenues for both leveraging the utility of, and developing on, cutting-edge computer vision and machine learning tools with minimal to no coding – namely, by only using intuitive language.

# 6    Acknowledgements

This work was supported by The Vallee Foundation Scholar award to MWM. We thank NeuroX at EPFL for travel support. We thank members of the M.W.Mathis Lab and A.Mathis Group at EPFL for feedback, A. Iqbal for computing animal-object interactions used in Appendix Figure 6, St. Schneider and Kinematik AI for assistance and hosting our demo, and the alpha testers. **COI:** MWM is a co-founder and equity holder in Kinematik AI. **Author Contributions:** Conceptualization: SY, MWM; Methodology & Software: SY, JL, MWM, MZ, AM; Writing: MWM, SY; Writing-Editing: JL, AM; Visualization: MWM, SY, MZ; Funding acquisition: MWM.

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

## Appendix

## System Prompts

Overall, AmadeusGPT consists of three GPT3.5 or GPT4 instances with different system prompts, the pretrained computer vision models (e.g., SAM and SuperAnimal) and our API interface and implementation. For the pretrained computer vision models (SAM, SuperAnimal), training details are provided in the original works, and we did not fine-tune them. For GPT3.5, the training details were not released by OpenAI at the time of this publication, and we also did not fine-tune any GPT model.

Below are the **core API system prompts** used to create AmadeusGPT; related to Main Section 3.2.

```
system_prompt = f"Your name is AmadeusGPT and you are helping our users by first
↪   understanding my API docs: \n{interface_str}\n{cls.behavior_modules_str}\n"
system_prompt += "Now that you understand my API docs, If user asks things that are
↪   achievable by using my APIs, write and only write a function named task_program()
↪   with return values. If not achieveable by API docs, explain why. Think step by step
↪   when generating code. \n "
system_prompt += "Rule 1: (1) Do not access attriibutes of objects, unless attributes are
↪   written in the Arributes part of function docs. (2) Do not call any functions or
↪   object methods unless they are written in Methods part in the func docs. \n"
system_prompt += "Rule 2: Pay attention to whether the user means for single animal or
↪   multiple animals.\n"
system_prompt += "Rule 3: Never call instance functions of matplotlib objects such as ax
↪   and plt.\n"
system_prompt += "Rule 4: Do not write any comments in the function.\n"
system_prompt += "Rule 5: Must pay attention to the typing for the parameters and returns
↪   of functions when writing code.\n"
system_prompt += "Rule 6: When generating events, pay attention to whether it is
↪   simultaneous or sequential from the user prompt. For example, events describing
↪   multiple bodyparts for one animal must be simultaneous events. Animal moving from one
↪   place to the other must be sequential events. \n"
```

Below are the Rephraser system prompts used to counter domain shift in the user prompt style; related to Main Section 3.2.

```
system_prompt = f"Your job is to help translate queries into more queries that follow my
↪   rules and examples.  If you do not know how to translate, just repeat what you are
↪   asked and never say sorry. \n"
system_prompt += "Rule 1: Make sure you only answer with a short converted version. No
↪   explanation or other formatting.\n"
system_prompt += "Rule 2: Make sure you do not change bodypart names and do not change
↪   name of the behaviors. Name of the behaviors might be enclosed by <||>.\n"
system_prompt += "Rule 3: Do not change temporal conjunctions when rephrasing. For
↪   example, 'and then' cannot be replaced by 'and'.\n"
system_prompt += (
    "Rule 4: Do not change the semantic meaning of the query. Do not remove important
    ↪   details.\n"
)
system_prompt += "Rule 5: Do not rephrase adjective words such as shortest event or
↪   longest distance.\n"
system_prompt += "Rule 6: when a spatial relation is used as adjcetive such as entering
↪   from right, add a adjcetive word simultanesouly such as entering from right
↪   simultaneously.\n"
system_prompt += "Rule 7: Do not rephrase terms with special meaning such as cebra, umap,
↪   or deeplabcut.\n"
system_prompt += "Rule 8: If you do not know how to rephrase it, just copy the
↪   instruction and write back.\n"
system_prompt += "Rule 9: Keep the adjectives for words like head angle or closest
↪   distance or relative speed. They should not be rephrased such that head angle is
↪   confused as angle or relative_speed confused as speed\n or distance confused as
↪   closest distance\n"
system_prompt += "Rule 10: Rephrase specialized animal names such as mouse, mice to
↪   animal, animals.\n"
```

```
system_prompt += f"Example 1: How much time the animal spends in ROI? -> Give me the
↪   amount of time animal spends in ROI and the events where the animal overlaps ROI.\n"
system_prompt += f"Example 2: How much time the animal moves faster than 3? -> Give me
↪   events where the animal moves faster than 3.\n"
system_prompt += f"Example 3: What is the duration of time animal is outside ROI? -> Get
↪   events where the animal is outside ROI.\n"
system_prompt += f"Example 4: What is distance travelled in ROI -> Get me distance
↪   travelled of the animal overlaps ROI.\n"
system_prompt += f"Example 5: Get object ethogram -> Plot ethogram for animal overlapping
↪   all objects.\n"
system_prompt += f"Example 6: Give me trajectories of the animal -> Plot me trajectories
↪   of the animal.\n"
system_prompt += f"Example 7: The animal enters object 1 and then enters object 2 -> The
↪   animal enters object 1 and then enters object.\n"
system_prompt += f"Example 8: Define <|watching|> as a social behavior where distance
↪   between animals is less than 260 and distance larger than 50 and angle between
↪   animals \
less than 15. Get masks for watching. -> Define <|watching|> as a social behavior where
↪   distance between animals is less than 260 and distance larger than 50 and angle
↪   between animals \
less than 15. Get events where watching happens.\n"
system_prompt += f"Example 10: plot object ethogram -> plot object ethogram.\n"
system_prompt += f"Example 11: define <|bla|> as a behavior where the animal's bodypart
↪   tail_base moves faster than 3 while animal's bodypart nose moves faster than 4, when
↪   does the animal do bla? -> define <|bla|> as a behavior where the animal's tail_base
↪   moves faster than 3 while animal's nose moves faster than 4. Give events where animal
↪   does bla.\n"
system_prompt += f"Example 12: Give me frames where the animal is on object 17 -> Give me
↪   events where the animal is on object 17.\n"
system_prompt += f"Example 13: Give me 3d cebra embedding -> Give me 3d cebra embedding
↪   and plot the embedding.\n"
system_prompt += f"Example 14: plot trajectory when the animal is on object 1 -> plot
↪   trajectory for events where the animal overlaps object 1.\n"
system_prompt += f"Example 15: Help me define the behavior freeze - > Help me define the
↪   behavior freeze.\n"
system_prompt += f"Example 16: Where is the animal in the first half of the video? ->
↪   Where is the animal in the first half of the frames?\n"
system_prompt += f"Example 17: When is the animal above object 3? -> Give me events where
↪   animal is above object 3.\n"
system_prompt += f"Example 18: When is the closest distance among animals is less than 3?
↪   - > Give me events where the closest_distance among animals is less than 3.\n"
system_prompt += f"Example 19: How much time does the mouse spend on roi0? -> Give me
↪   amount of time animal spends on roi0 and events where the animal overlap with ROI0
↪   \n "
system_prompt += f"Before rephrasing, keep in mind all those rules need to be followed."
```

Below are the self-correction system prompts used to fix common runtime errors; ; related to Main Section 3.2.

```
system_prompt = """ Your job is to correct a code that raised errors. You will be given
↪   the user query the code was written for, the code, the error message and the
↪   diagnosis for why the error happens
"""
system_prompt+= "There are empirical rules that can help you debug:\n"
system_prompt+= f"Rule 1: If the code used a bodypart that does not exist, replace it
↪   with one that is semantically close from supported bodyparts. \n"
```

## Dual memory mechanism, reload and relaunch example

Related to Main Section 3.3, we discuss how short-term memory can be supplemented by long-term memory to restore chat history. AmadeusGPT stores information including chat histories in both short-term and long-term memory as well as segmentation masks and relation look-up tables. As they are all implemented as Python objects, it is easy to serialize them into disk and deserialize them from the disk. More specifically, we support two special prompts called "save" and "load" respectively.

Those two special prompts can save the state of AmadeusGPT before exiting and load the state of AmadeusGPT after relaunching.

**Query:** "save"

**AmadeusGPT Output:**

```
Saving state into state.pickle
```

**Query:** "load"

**AmadeusGPT Output:**

```
Loaded state.pickle
```

## API core behavioral modules

This is part of the core API docs that are always sent to AmadeusGPT when queried; related to Main Section 3.4.

```python
class AnimalBehaviorAnalysis:
    """
    Methods
    -------
    get_object_names() -> List[str]
        get names of all objects
    """
    def animals_state_events(self,
                             state_type,
                             comparison,
                             bodyparts = ['all'],
                             ):
        """
        Parameters
        ----------
        state_type: str
            Must be one of 'speed', 'acceleration'
        comparison: str
            Must be a comparison operator followed by a number like <50,
        Examples
        --------
        >>>  # when is the animal moving faster than 3 pixels across frames?
        >>>  def task_program():
        >>>      behavior_analysis = AnimalBehaviorAnalysis()
        >>>      animal_faster_than_3_events =
↪ behavior_analysis.animals_state_events('speed', '>3')
        >>>      return animal_faster_than_3_events
        """
        return animals_state_events(state_type, comparison)

    def superanimal_video_inference(self) -> None:
        """
        Examples
        --------
        >>> # extract keypoints (aka pose) from the video file
        >>> def task_program():
            >>>      behavior_analysis = AnimalBehaviorAnalysis()
        >>>      behavior_analysis.superanimal_video_inference()
        """
        return superanimal_video_inference()

    def animals_object_events(self,
```

```python
                                    object_name: str,
                                    relation_query,
                                    comparison = None,
                                    negate = False,
                                    bodyparts: List[str] = ['all'],
                                    min_window = 0,
                                    ) -> EventDict:
        """
        object_name : str. Name of the object
        relation_query: str. Must be one of 'to_left', 'to_right', 'to_below',
↪  'to_above', 'overlap', 'distance', 'angle', 'orientation'
        comparison : str, Must be a comparison operator followed by a number like <50,
↪  optional
        bodyparts: List[str], optional
        min_window: min length of the event to include
        max_window: max length of the event to include
        negate: bool, default false
            whether to negate the spatial events. For example, if negate is set True,
↪  inside roi would be outside roi
        Returns:
        -------
        EventDict
        Examples
        --------
        >>> # find where the animal is to the left of object 6
        >>> def task_program():
        >>>     behavior_analysis = AnimalBehaviorAnalysis()
        >>>     left_to_object6_events = behavior_analysis.animals_object_events('6',
↪  'to_left',  bodyparts = ['all'])
        >>>     return left_to_object6_events
        >>> # how much time the animal spends in closed arm?
        >>> def task_program():
        >>>     behavior_analysis = AnimalBehaviorAnalysis()
        >>>     in_closed_arm_events = behavior_analysis.animals_object_events('closed
↪  arm', 'overlap', bodyparts = ['all'], negate = False)
        >>>     return in_closed_arm_events
        >>> # get events where animals's distance to animal0 is less than 30
        >>> def task_program():
        >>>     behavior_analysis = AnimalBehaviorAnalysis()
        >>>     distance_social_events =
↪  behavior_analysis.animals_object_events('animal0', 'disntace', comparison = '<30',
↪  bodyparts = ['nose'])
        >>>     return distance_social_events
        """
        return animals_object_events(object_name, relation_query)

    def plot_trajectory(self, bodyparts: List[str], events: Union[Dict[str, dict],dict] =
↪  None, **kwargs):
        """
        Parameters
        ----------
        bodyparts : List[str]
        events: Union[Dict[str, dict],dict], optional
            The type must be either dict or dictionary of dict
        kwargs : parameters for plt.scatter
        Returns:
        -------
        Tuple[plt.Figure, plt.Axes]
            tuple of figure and axes
        -------
        Examples
        --------
        >>> # plot trajectory of the animal.
```

```
>>> def task_program():
>>>     behavior_analysis = AnimalBehaviorAnalysis()
>>>     fig, ax = behavior_analysis.plot_trajectory(["all"])
>>>     return fig, ax
>>> # plot trajectory of the nose of the animal with the event that animal
↪  overlaps with object 6
>>> def task_program():
>>>     behavior_analysis = AnimalBehaviorAnalysis()
>>>     overlap_6_events = behavior_analysis.animals_object_events('6',
↪  'overlap', bodyparts = ['all'])
>>>     fig, ax = behavior_analysis.plot_trajectory(["nose"], events =
↪  overlap_6_events)
>>>     return fig, ax
"""
    return plot_trajectory(bodyparts, events)

# do not call animals_social_events if there is no multiple animals or social events
def animals_social_events(self,
                          relation_query_list,
                          comparison_list,
                          animal_state_relation_query_list = [],
                          animal_state_comparison_list = [],
                          bodyparts = ['all'],
                          otheranimal_bodyparts = ['all'],
                          min_window = 11,
                          pixels_per_cm = 8,
                          smooth_window_size = 5):
    """
        The function is specifically for capturing multiple animals social events
    Parameters
    ----------
    relation_query_list: List[str]
    list of relation query for animals-animals relationship. Must be chosen from
↪  ['to_left', 'to_right', 'to_below', 'to_above', 'overlap', 'distance', 'orientation',
↪  'closest_distance', 'angle'].
    comparison_list: List[str]
    list of comparison operator such as '==', '<', '>', '<=', '>='.
    animal_state_relation_query_list: List[str]
    list of relation query for state of the animal that interacts with other animals.
↪  Must be chosen from ['speed', 'acceleration', 'confidence'].
    animal_state_comparison_list: List[str]:
    list of comparison operator such as '==', '<', '>', '<=', '>='.
    bodyparts: List[str]
    list of bodyparts for the main animal
    otheranimal_bodyparts: list[str]
    list of bodyparts for the other animal
    min_window: int, optional
    Only include events that are longer than min_window
    pixels_per_cm: int, optional
    how many pixels for 1 centimer
    smooth_window_size: int, optional
    smooth window size for smoothing the events.
    Examples
    --------
    >>> # Define <|chases|> as a social behavior where distance between animals are
↪  less than 100, one animal is in front of the other animal and the chasing animal has
↪  speed faster than 3. Get chases.
    >>> def task_program():
    >>>     chase_events = behavior_analysis.animals_social_events(['distance',
↪  'orientation'],
    >>>         [f'< 100', f'=={Orientation.FRONT}'],
    >>>         animal_state_relation_query_list = ['speed'],
```

```
>>>            animal_state_comparison_list=['>=3'],
>>>            bodyparts = ['all'],
>>>            otheranimal_bodyparts = ['all'])
>>>        return chase_events
"""
        return animals_social_events(relation_query_list, comparison_list)

class Orientation(IntEnum):
    """
    Attributes
    ----------
    FRONT
    BACK
    LEFT
    RIGHT
    """
class Event:
    @classmethod
    def add_simultaneous_events(cls, *events_list: List[dict])-> dict:
        """
        Parameters
        ----------
        events_list: List[dict]
        Returns
        -------
        EventDict
        Examples
        --------
        >>> get events for the animal's nose in the roi and the animal's tail base not in
↪   the roi
        >>> def task_program():
        >>>     behavior_analysis = AnimalBehaviorAnalysis()
        >>>     nose_in_roi_events =  behavior_analysis.animals_object_events('ROI',
↪   'overlap', bodyparts = ['nose'], negate = False)
        >>>     tail_base_not_in_roi_events =
↪   behavior_analysis.animals_object_events('ROI', 'overlap', bodyparts = ['tail base'],
↪   negate = True)
        >>>     return Event.add_simultaneous_events(nose_in_roi_events,
↪   tail_base_not_in_roi_events)
        """
        return add_simultaneous_events(events_list)
    @classmethod
    def add_sequential_events(cls,
                              *events_list: List[dict],
                              continuous = False)-> dict:
        """
        Keywords such as "then", "later" might suggest it is a sequential event
        Parameters
        ----------
        events_list: List[dict]
        Returns
        -------
        EventDict
        Examples
        --------
        >>> # Get events where the animal enters object 6 from the bottom.
        >>> def task_program():
        >>>     behavior_analysis = AnimalBehaviorAnalysis()
        >>>     bottom_to_object_6_events =
↪   behavior_analysis.animals_object_events('6','to_below', bodyparts = ['all'])
        >>>     enter_object_6_events = behavior_analysis.enter_object('6', bodyparts =
↪   ['all'])
        >>>     enter_object_6_from_bottom_events =
↪   Event.add_sequential_events(bottom_to_object_6_events, enter_object_6_events)
```

```
    >>>        return enter_object_6_from_bottom_events
    >>> # Find events where animal leaves object 6
    >>> def task_program():
    >>>        behavior_analysis = AnimalBehaviorAnalysis()
    >>>        leave_object_6_events = behavior_analysis.leave_object('6', bodyparts =
↪ ['all'])
    >>>        return leave_object_6_events
    >>> # Find events where the animal leaves object 6 and then enters object 3
    >>> def task_program():
    >>>        behavior_analysis = AnimalBehaviorAnalysis()
    >>>        leave_object_6_events = behavior_analysis.leave_object('6', bodyparts =
↪ ['all'])
    >>>        enter_object_3_events = behavior_analysis.enter_object('3', bodyparts =
↪ ['all'])
    >>>        leave_object_6_and_enter_object_3_events =
↪ Event.add_sequential_events(leave_object_6_events, enter_object_3_events)
    >>>        return leave_object_6_and_enter_object_3_events
    >>> # find events where the animal moves from left of object 6 to right of object
↪ 6.
    >>> def task_program():
    >>>        behavior_analysis = AnimalBehaviorAnalysis()
    >>>        left_to_object_6_events =
↪ behavior_analysis.animals_object_events('6','to_left', bodyparts = ['all'])
    >>>        right_to_object_6_events =
↪ behavior_analysis.animals_object_events('6','to_right', bodyparts = ['all'])
    >>>        left_to_right_events =
↪ Event.add_sequential_events(left_to_object_6_events, right_to_object_6_events)
    >>>        return left_to_right_events
    """
    return add_sequential_events(events_list)

class EventDict(dict):
    def sort(self):
        """
        Sort the eventdict
        Return
        ------
        EventDict
        """
        return sort()
```

## Integration of behavioral modules (non-exhaustive)

As described in the Main Section 3.5 and Main Figure 2, AmadeusGPT can use integration modules
for task-specific behavior analysis, and implements dynamic loading by converting API documents
into text embedding vectors and caching them in RAM or disk, allowing users to retrieve the most
relevant modules by cosine similarity using their prompt as a query. Namely, the user could use these
modules to integrate functionality. Or, they can write simple modules and contribute them to the
AmadeusGPT codebase for other users to leverage as well. Below we provide examples of integration
with the newly introduced dimensionality reduction tool called cebra [35], and UMAP [34]. In short,
to integrate into the codebase the user should write a minimal working example in the style of the
core API docs. Then, after dynamic loading, queries relating to UMAP or cebra (as shown here)
would generate the following code (below) and related output plots (Figure 6).

### CEBRA

CEBRA is a nonlinear dimensionality reduction tool for joint modeling of neural data and behavioral
labels [35], which might be of interest to users of AmadeusGPT. In Appendix Figure 6 we show a
3D CEBRA embedding of mouse poses (aligned) colored by when it overlaps with different objects.
Note that the CEBRA embedding in Figure 6 is on a 3D n-sphere that is equivalent to a 2D Euclidean

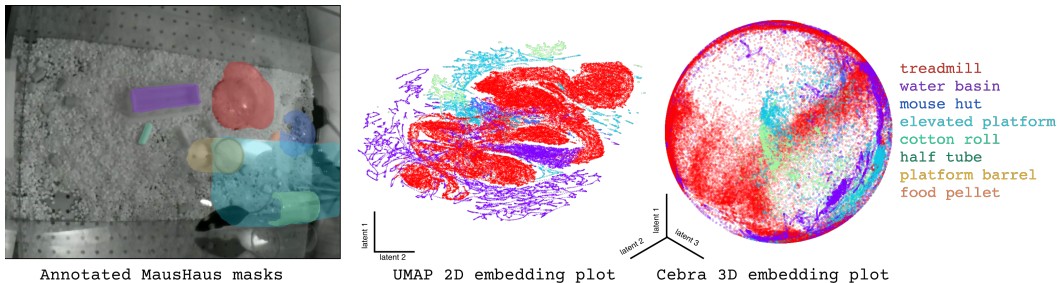

| Annotated MausHaus masks | UMAP 2D embedding plot | Cebra 3D embedding plot |

treadmill
water basin
mouse hut
elevated platform
cotton roll
half tube
platform barrel
food pellet

Figure 6: **Integration examples.** Left to right: MausHaus masks with human annotated names, examples of UMAP and cebra integration modules, plots colored as in annotations. See below for user prompts to produce the plots.

space [35]. Below the user query is shown below to produce images in Figure 6, along with the related API docs.

**Query:** "get 3D embeddings using cebra and plot the embedding."

```python
def compute_embedding_with_cebra(n_dimension = 3,
                                 max_iterations=100):
    """
    Examples
    --------
    >>> # create a 3 dimensional embedding with cebra
    >>> def task_program():
    >>>     behavior_analysis = AnimalBehaviorAnalysis()
    >>>     embedding = behavior_analysis.compute_embedding_with_cebra(n_dimension = 3)
    >>>     return embedding
    """
    return compute_embedding_with_cebra(n_dimension = n_dimension,
    ↪   max_iterations=max_iterations)
```

## UMAP

UMAP [34] is a popular nonlinear dimensionality reduction tool that is also core to several behavioral analysis tools such as B-SOID [46]. In Appendix Figure 6 we show 2D UMAP embedding of poses (aligned) when it overlaps with different objects.

**Query:** "Get 2D embeddings using umap and plot the embedding"

```python
def compute_embedding_with_umap(n_dimension = 3):
    """
    Examples
    --------
    >>> # create a 3 dimensional embedding  umap
    >>> def task_program(features):
    >>>     behavior_analysis = AnimalBehaviorAnalysis()
    >>>     embedding = behavior_analysis.compute_embedding_with_umap(n_dimension = 3)
    >>>     return embedding
    """
    return compute_embedding_with_umap(n_dimension = n_dimension)
```

### Global spatio-temporal behavioral analyses

Many core behavioral analysis tools rely on computing metrics based on kinematic or movement analysis. For example, time spent locomoting, time in an ROI, or computing the distance traveled within a specific area is highly popular in behavioral neuroscience research. Thus, inspired by DLC2Kinematics [47], we show API docs to integrate two such common integration modules:

```python
def calculate_distance_travelled(self, events):
    """
    Parameters
    ----------
```

```
    events: EventDict
    Return
    ------
    int
    Examples
    --------
    >>> # what is the distance travelled in closed arm?
    >>> def task_program():
    >>>     behavior_analysis = AnimalBehaviorAnalysis()
    >>>     roi_events = behavior_analysis.animals_object_events('closed arm', 'overlap',
↪   bodyparts = ['all'])
    >>>     distance_in_roi = behavior_analysis.calculate_distance_travelled(roi_events)
    >>>     return distance_in_roi
    """
    return calculate_distance_travelled(events)

def plot_object_ethogram(self, object_names):
    """
    Parameters
    ----------
    object_names: List[str]
    list of object names
    Return:
    -------
    Tuple[plt.Figure, plt.Axes]
        tuple of figure and axes
    Examples
    --------
    >>> # plot ethogram for animal overlapping all objects
    >>> def task_program():
    >>>     behavior_analysis = AnimalBehaviorAnalysis()
    >>>     object_names = behavior_analysis.get_object_names()
    >>>     fig, ax = behavior_analysis.plot_object_ethogram(object_names)
    >>>     return fig, ax
    """
    return plot_object_ethogram(object_names)
```

## Additional Experiments: behavioral analysis examples

Here, as related to Main Section 4, we provide more example queries and responses from
AmadeusGPT to support Main Figure 5a (EPM), to Main Figure 5b (MausHaus), and to Main
Figure 5c (MABe) and results in Table 1.

### Extended Results on EPM with AmadeusGPT

We show additional user queries and generated code from AmadeusGPT related to the EPM data
presented in the Main Figure 5a.

### Extended Results on MausHaus with AmadeusGPT

We show the user queries and generated code from AmadeusGPT that produce the images shown in
the Main Figure 5b.

**Query:** "Give me the trajectories of the animal entering object 18 from right"

**AmadeusGPT Code Output:**

```
def task_program():
    behavior_analysis = AnimalBehaviorAnalysis()
    right_to_object_18_events = behavior_analysis.animals_object_events('18','to_right',
↪   bodyparts = ['all'])
    enter_object_18_events = behavior_analysis.enter_object('18', bodyparts = ['all'])
    enter_object_18_from_right_events =
↪   Event.add_sequential_events(right_to_object_18_events, enter_object_18_events)
```

👤 **Query:**

"Define <|head dips|> as a behavior where the animal's bodyparts mouse_center and at the same time neck in the roi and its head_midpoint outside roi. Get head dips."

🪓 **Amadeus Code Outputs:**

```python
def task_program():
    behavior_analysis = AnimalBehaviorAnalysis()
    mouse_center_neck_in_roi_events = behavior_analysis.animals_object_events( \
            'ROI', 'overlap', bodyparts=['mouse_center', 'neck'], negate=False)
    head_midpoint_not_in_roi_events = behavior_analysis.animals_object_events( \
            'ROI', 'overlap', bodyparts=['head_midpoint'], negate=True)
    head_dips_events = Event.add_simultaneous_events( \
            mouse_center_neck_in_roi_events, head_midpoint_not_in_roi_events)
    return head_dips_events
```

🪓 **Amadeus Chat Outputs (Execution):**

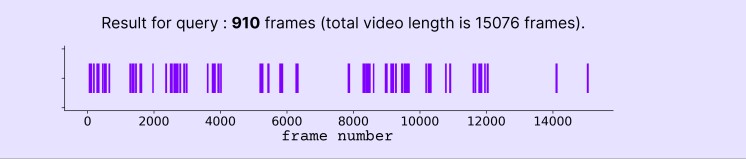

👤 **Query:**

"When is the animal moving faster than 8 pixels per frame?"

🪓 **Amadeus Code Outputs:**

```python
def task_program():
    behavior_analysis = AnimalBehaviorAnalysis()
    animal_faster_than_8_events = behavior_analysis.animals_state_events( \
        'speed', '>8')
    return animal_faster_than_8_events
```

🪓 **Amadeus Chat Outputs (Execution):**

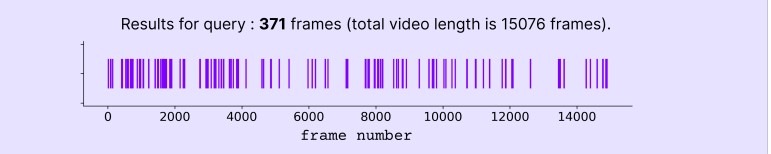

👤 **Query:**

"Define <|freezing|> as a behavior where the animal is slower than 0.5. When is the mouse freezing?"

🪓 **Amadeus Code Outputs:**

```python
def task_program():
    behavior_analysis = AnimalBehaviorAnalysis()
    freezing_events = behavior_analysis.animals_state_events('speed', '<0.5')
    return freezing_events
```

🪓 **Amadeus Chat Outputs (Execution):**

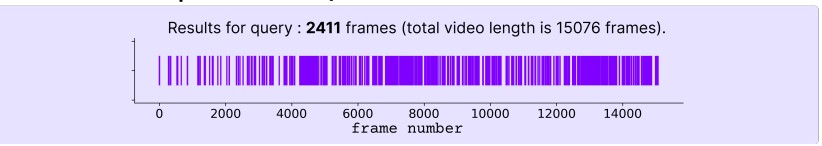

Figure 7: **Extended Results on classical behavioral task (EPM) with AmadeusGPT**. **(a)** Using the EPM dataset [13], we show additional queries and responses from AmadeusGPT. The raster plot is an ethogram where there is a tick for every related event in time, i.e., the mouse is in the closed arm or in the open arm.

```python
    fig, ax = behavior_analysis.plot_trajectory(["all"], events =
↪    enter_object_18_from_right_events)
    return fig, ax
```

**Query:** "plot trajectories of the animal for the shortest event where it enters object 18 and then its neck enters object 17"

**AmadeusGPT Code Output:**

```python
def task_program():
    behavior_analysis = AnimalBehaviorAnalysis()
```

```
enter_object_18_events = behavior_analysis.enter_object('18', bodyparts=['all'])
neck_enter_object_17_events = behavior_analysis.enter_object('17',
↪   bodyparts=['neck'])
enter_object_18_and_neck_enter_object_17_events =
↪   Event.add_sequential_events(enter_object_18_events, neck_enter_object_17_events)
fig, ax = behavior_analysis.plot_trajectory(bodyparts=['all'],
↪   events=enter_object_18_and_neck_enter_object_17_events)
return fig, ax
```

**MABe Mouse Triplet Benchmark, extended details**

In the MABe 2022 Mouse Triplet Benchmark, participants are given a dataset of tracking data of triplets of socially interacting mice and asked to submit a frame-by-frame representation of the dataset. Here are the following user prompts and AmadeusGPT code outputs of the sub-tasks we considered and the generated code by AmadeusGPT.

**Query:** "Define <|chases|> as a social behavior where the closest distance between animals is less than 40 pixels and the chased animal is in front of the chasing animal and the chasing animal has to travel faster than 3.4 pixels per frame. The smooth window size is 25 and min window is 30. Get chases."

**AmadeusGPT Code Output:**

```
def task_program():
    behavior_analysis = AnimalBehaviorAnalysis()
    chase_events = behavior_analysis.animals_social_events(['closest_distance',
    'orientation'],
    ['<40', '=='+str(Orientation.FRONT)],
    animal_state_relation_query_list=['speed'],
    animal_state_comparison_list=['>3.4'],
    bodyparts=['all'],
    otheranimal_bodyparts=['all'],
    smooth_window_size=25,
    min_window=30)
    return chase_events
```

**Query:** "Define <|oral genital contact|> as a social behavior where closest distance between this animal's "nose" and other animal's bodyparts "tail base" is less than 15, get the events for oral genital contact."

**AmadeusGPT Code Output:**

```
def task_program():
    behavior_analysis = AnimalBehaviorAnalysis()
    oral_genital_contact_events =
    ↪   behavior_analysis.animals_social_events(['closest_distance'],
      ['< 15'],
      bodyparts = ['nose'],
      otheranimal_bodyparts = ['tail base'])
    return oral_genital_contact_events
```

**Query:** " Define <|oral oral contact|> as a social behavior where closest distance between this animal's "nose" and other animal's "nose" is less than 15, get events for oral oral contact."

**AmadeusGPT Code Output:**

```
def task_program():
    behavior_analysis = AnimalBehaviorAnalysis()
    oral_oral_contact_events =
    ↪   behavior_analysis.animals_social_events(['closest_distance'],
    ['< 15'],
    bodyparts = ['nose'],
```

```
        otheranimal_bodyparts = ['nose'])
    return oral_oral_contact_events
```

**Query:** "Define <|approaches|> as a social behavior where the mice move from at least 40 pixels apart sequentially to less than 8 pixels apart. Get approaches events."

**AmadeusGPT Code Output:**

```
def task_program():
    behavior_analysis = AnimalBehaviorAnalysis()
    distance_events = behavior_analysis.animals_social_events(['distance'], ['>40'])
    close_distance_events = behavior_analysis.animals_social_events(['distance'], ['<8'])
    approaches_events = Event.add_sequential_events(distance_events,
    ↪  close_distance_events)
    return approaches_events
```

**Query:** "Define <|huddles|> as a social behavior where the distance between animals is less than 50, and the relative speed between two animals is less than 4 pixels per frame. The smooth window size is 61 and min window is 75. Get huddles."

**AmadeusGPT Code Output:**

```
def task_program():
    behavior_analysis = AnimalBehaviorAnalysis()
    huddle_events = behavior_analysis.animals_social_events(['distance', 'speed'],
    ['<50', '<4'],
    bodyparts=['all'],
    min_window=75,
    smooth_window_size=61)
    return huddle_events
```

**Query:** "Define <|contact|> as a social behavior where the closest distance between animals is less than 12. The smooth window size is 11 and min window is 5. Get contact."

**AmadeusGPT Code Output:**

```
def task_program():
    behavior_analysis = AnimalBehaviorAnalysis()
    contact_events = behavior_analysis.animals_social_events(['closest_distance'],
    ↪  ['<12'], smooth_window_size=11, min_window=5)
    return contact_events
```

**Query:** "Define <|close|> as a social behavior where the closest distance between animals is less than 24. The min window is 5. Get close."

**AmadeusGPT Code Output:**

```
def task_program():
    behavior_analysis = AnimalBehaviorAnalysis()
    close_events = behavior_analysis.animals_social_events(['closest_distance'], ['<24'],
    ↪  min_window=5)
    return close_events
```

**Query:** "Define <|watching|> as a social behavior where distance between animals is less than 260 and larger than 50 and angle between animals is less than 15. The min window is 100 and smooth window is 15. Get watching."

**AmadeusGPT Code Output:**

```
def task_program():
    behavior_analysis = AnimalBehaviorAnalysis()
    watching_events = behavior_analysis.animals_social_events(['distance', 'distance',
    ↪  'angle'],
    ['<260', '>50', '<15'],
    bodyparts=['all'],
```

```
    min_window=100,
    smooth_window_size=15)
    return watching_events
```

**Query:** "Define <|oral ear contact|> as a social behavior where the closest distance between animal's nose and other animals' left ear and right ear is less than 10. The smooth window size is 5 and min window is 15. Get oral ear contact."

**AmadeusGPT Code Output:**

```
def task_program():
    behavior_analysis = AnimalBehaviorAnalysis()
    oral_ear_contact_events =
    ↪   behavior_analysis.animals_social_events(['closest_distance',
    ↪   'closest_distance'],
    ['<10', '<10'],
    bodyparts=['nose'],
    otheranimal_bodyparts=['left ear', 'right ear'],
    min_window=15,
    smooth_window_size=5)
    return oral_ear_contact_events
```

## Prompt robustness testing

Related to Main Section 4, we show the full base questions and generated permutations we tested to gauge robustness.

### The input, base out of distribution (OOD) questions

```
"""
(1) define <|head dips|> as a behavior where the animal's bodypart mouse_center bodypart
↪   neck in the roi and its bodypart head_midpoint outside roi
(2) When is the animal moving faster than 8?
(3) define <|freezing|> as a behavior where the animal is slower than 0.5. When is the
↪   mouse freezing?
(4) How much time the animal spends in the closed arm?
(5) How much time the animal spends outside the closed arm?
"""
```

### The 25 generated OOD questions

```
"""
(1) Dips refer to a behavior in which an animal's bodypart, specifically the mouse_center
↪   and neck, are within the ROI while the head_midpoint is outside the ROI.
(2) When an animal's mouse_center and neck are inside the ROI but the head_midpoint is
↪   outside, this is known as dips.
(3) Dips are characterized by the presence of an animal's mouse_center and neck within
↪   the ROI, while the head_midpoint is located outside the ROI.
(4) The behavior known as dips occurs when an animal's mouse_center and neck are within
↪   the ROI, but the head_midpoint is outside of it.
(5) In animal behavior, dips are defined as the presence of the mouse_center and neck
↪   within the ROI, while the head_midpoint is situated outside of it.
(6) At what point does the animal exceed a speed of 8?
(7) When does the animal go faster than 8?
(8) At what moment does the animal surpass a speed of 8?
(9) When is the animal traveling at a speed greater than 8?
(10) At what time does the animal go beyond a speed of 8?
(11) At what point can we say that a mouse is freezing? I define freezing as a behavior
↪   where the animal moves slower than 0.5.
(12) When does a mouse exhibit freezing behavior? Freezing is defined as a state where
↪   the animal's movement is slower than 0.5.
(13) What is the definition of freezing in mice? Freezing is characterized by a movement
↪   speed of less than 0.5. When does a mouse exhibit this behavior?
```

```
(14) When can we say that a mouse is freezing? Freezing is a behavior where the animal's
↪   movement is slower than 0.5. What are the conditions that lead to this behavior?
(15) What does it mean when we say that a mouse is freezing? Freezing is a state where
↪   the animal moves slower than 0.5. When does this behavior typically occur in mice?
(16) What is the duration for which the animal remains in the enclosed arm?
(17) How long does the animal spend in the closed arm?
(18) What is the time frame for which the animal stays in the enclosed arm?
(19) What is the length of time that the animal spends in the closed arm?
(20) What is the period of time during which the animal remains in the enclosed arm?
(21) What is the duration for which the animal remains outside the closed arm?
(22) How long does the animal stay outside the closed arm?
(23) What is the amount of time the animal spends outside the closed arm?
(24) For how much time does the animal remain outside the closed arm?
(25) What is the length of time that the animal spends outside the closed arm?
"""
```

**Questions rephrased automatically by our Rephraser**

```
"""
(1) Define <|dips|> as a behavior where the animal's mouse_center and neck are in ROI and
↪   head_midpoint is outside ROI.
(2) Define dips as an event where the animal's mouse_center and neck are inside the ROI
↪   but the head_midpoint is outside.
(3) Analyze events where the animal's mouse_center and neck are in ROI and head_midpoint
↪   is outside ROI.
(4) Define <|dips|> as a behavior where the animal's mouse_center and neck are within
↪   ROI, but head_midpoint is outside.
(5) Define dips as the presence of mouse_center and neck within ROI and head_midpoint
↪   outside of it.
(6) Give me events where the animal exceeds a speed of 8.
(7) Give me events where the animal goes faster than 8.
(8) Give me events where the animal's speed is greater than 8.
(9) Give me events where the animal is traveling at a speed greater than 8.
(10) Give me events where the animal goes beyond a speed of 8.
(11) Define freezing as a behavior where the animal moves slower than 0.5. When does the
↪   mouse freeze?
(12) Give events where the mouse exhibits freezing behavior, defined as movement slower
↪   than 0.5.
(13) Define freezing as movement speed less than 0.5 in mice. Give events where a mouse
↪   exhibits freezing behavior.
(14) Define freezing as a behavior where the animal's movement is slower than 0.5. What
↪   are the conditions that lead to freezing in a mouse?
(15) What is freezing in mice? Give events where the mouse is freezing.
(16) Get events where the animal is in the enclosed arm and calculate their duration.
(17) Give me events where the animal is in the closed arm.
(18) Get events where the animal stays in the enclosed arm.
(19) Get events where the animal is in the closed arm.
(20) Get events where the animal remains in the enclosed arm.
(21) Get events where the animal is outside the closed arm.
(22) Get events where the animal is outside the closed arm and give me their duration.
(23) Get events where the animal is outside the closed arm.
(24) Give me events where the animal is outside the closed arm.
(25) Get events where the animal is outside the closed arm.
"""
```

**Code generated by AmadeusGPT without Rephraser**

```python
#Occurs 5 times
def task_program():
    behavior_analysis = AnimalBehaviorAnalysis()
    # Get events where mouse_center and neck are within the ROI
    dip_events = behavior_analysis.animals_object_events('ROI', 'overlap',
    ↪   bodyparts=['mouse_center', 'neck'], negate=False)
    # Get events where head_midpoint is outside the ROI
```

```python
    head_outside_roi_events = behavior_analysis.animals_object_events('ROI', 'overlap',
    ↪  bodyparts=['head_midpoint'], negate=True)
    # Combine the events using add_sequential_events()
    dip_with_head_outside_roi_events = Event.add_sequential_events(dip_events,
    ↪  head_outside_roi_events)
    return dip_with_head_outside_roi_events
#Occurs 5 times
def task_program():
    behavior_analysis = AnimalBehaviorAnalysis()
    # Get events where the animal's speed exceeds 8 pixels per frame
    speed_exceed_8_events = behavior_analysis.animals_state_events('speed', '>8')
    return speed_exceed_8_events
#Occurs 3 times
def task_program():
    behavior_analysis = AnimalBehaviorAnalysis()
    # Get events where the animal's speed is less than 0.5 pixels per frame
    freezing_events = behavior_analysis.animals_state_events('speed', '<0.5')
    return freezing_events
#Occurs 1 times
def task_program():
    return "Freezing in mice is defined as a state where the animal moves at a speed of
    ↪  less than 0.5."
#Occurs 1 times
def task_program():
    behavior_analysis = AnimalBehaviorAnalysis()
    # Get events where the animal is in the closed arm
    in_closed_arm_events = behavior_analysis.animals_object_events('closed arm',
    ↪  'overlap', bodyparts=['all'], negate=False)
    # Calculate the duration of each event
    durations = [event.duration() for event in in_closed_arm_events['animal0']]
    # Return the average duration
    return sum(durations) / len(durations) if durations else 0.0
#Occurs 2 times
def task_program():
    behavior_analysis = AnimalBehaviorAnalysis()
    # Get events where the animal is in the closed arm
    in_closed_arm_events = behavior_analysis.animals_object_events('closed arm',
    ↪  'overlap', bodyparts=['all'], negate=False)
    # Calculate the duration of each event
    durations = [event.duration() for event in in_closed_arm_events['animal0']]
    # Return the total duration
    return sum(durations) if durations else 0.0
#Occurs 1 times
def task_program():
    behavior_analysis = AnimalBehaviorAnalysis()
    # Get events where the animal is in the closed arm
    in_closed_arm_events = behavior_analysis.animals_object_events('closed arm',
    ↪  'overlap', bodyparts=['all'], negate=False)
    # Get the start and end times of each event
    start_times = [event.start_frame for event in in_closed_arm_events['animal0']]
    end_times = [event.end_frame for event in in_closed_arm_events['animal0']]
    # Return the time frame of the first event if it exists
    if start_times:
        return (start_times[0], end_times[0])
#Occurs 1 times
def task_program():
    behavior_analysis = AnimalBehaviorAnalysis()
    # Get events where the animal is in the closed arm
    in_closed_arm_events = behavior_analysis.animals_object_events('closed arm',
    ↪  'overlap', bodyparts=['all'], negate=False)
    # Get the start and end times of each event
    event_times = [(event.start_time(), event.end_time()) for event in
    ↪  in_closed_arm_events['animal0']]
    # Return the event times
    return event_times if event_times else []
```

```
#Occurs 5 times
def task_program():
    behavior_analysis = AnimalBehaviorAnalysis()
    # Get events where the animal is not in the closed arm
    not_in_closed_arm_events = behavior_analysis.animals_object_events('closed arm',
    ↪    'overlap', bodyparts=['all'], negate=True)
    # Calculate the duration of each event
    durations = [event.duration() for event in not_in_closed_arm_events['animal0']]
    # Return the total duration
    return sum(durations) if durations else 0.0
```

**Code generated by AmadeusGPT with Rephraser**

```
#Occurs 4 times
def task_program():
    behavior_analysis = AnimalBehaviorAnalysis()
    mouse_center_in_roi_events = behavior_analysis.animals_object_events('ROI',
    ↪    'overlap', bodyparts=['mouse_center'], negate=False)
    neck_in_roi_events = behavior_analysis.animals_object_events('ROI', 'overlap',
    ↪    bodyparts=['neck'], negate=False)
    head_midpoint_not_in_roi_events = behavior_analysis.animals_object_events('ROI',
    ↪    'overlap', bodyparts=['head_midpoint'], negate=True)
    dips_events = Event.add_simultaneous_events(mouse_center_in_roi_events,
    ↪    neck_in_roi_events, head_midpoint_not_in_roi_events)
    return dips_events
#Occurs 1 times
def task_program():
    behavior_analysis = AnimalBehaviorAnalysis()
    mouse_center_in_roi_events = behavior_analysis.animals_object_events('ROI',
    ↪    'overlap', bodyparts=['mouse_center'], negate=False)
    neck_in_roi_events = behavior_analysis.animals_object_events('ROI', 'overlap',
    ↪    bodyparts=['neck'], negate=False)
    head_midpoint_not_in_roi_events = behavior_analysis.animals_object_events('ROI',
    ↪    'overlap', bodyparts=['head_midpoint'], negate=True)
    dips_events = Event.add_simultaneous_events(mouse_center_in_roi_events,
    ↪    neck_in_roi_events, head_midpoint_not_in_roi_events)
    fig, ax = behavior_analysis.plot_trajectory(bodyparts=['all'], events=dips_events)
    return fig, ax
#Occurs 5 times
def task_program():
    behavior_analysis = AnimalBehaviorAnalysis()
    faster_than_8_events = behavior_analysis.animals_state_events('speed', '>8')
    return faster_than_8_events
#Occurs 4 times
def task_program():
    behavior_analysis = AnimalBehaviorAnalysis()
    freezing_events = behavior_analysis.animals_state_events('speed', '<0.5')
    return freezing_events
#Occurs 1 times
def task_program():
    behavior_analysis = AnimalBehaviorAnalysis()
    in_closed_arm_events = behavior_analysis.animals_object_events('closed arm',
    ↪    'overlap', bodyparts = ['all'], negate = False)
    in_closed_arm_duration = in_closed_arm_events.duration()
    return in_closed_arm_duration
#Occurs 4 times
def task_program():
    behavior_analysis = AnimalBehaviorAnalysis()
    in_closed_arm_events = behavior_analysis.animals_object_events('closed arm',
    ↪    'overlap', bodyparts=['all'])
    return in_closed_arm_events
#Occurs 4 times
def task_program():
    behavior_analysis = AnimalBehaviorAnalysis()
```

```
    outside_closed_arm_events = behavior_analysis.animals_object_events('closed arm',
    ↪ 'overlap', bodyparts=['all'], negate=True)
    return outside_closed_arm_events
#Occurs 1 times
def task_program():
    behavior_analysis = AnimalBehaviorAnalysis()
    outside_closed_arm_events = behavior_analysis.animals_object_events('closed arm',
    ↪ 'overlap', bodyparts=['all'], negate=True)
    outside_closed_arm_duration = {animal: [event.duration() for event in events] for
    ↪ animal, events in outside_closed_arm_events.items()}
    return outside_closed_arm_duration
```

**Additional stress-testing with naive users and GPT4**

We additionally stress-tested the system with naive users. We released a fully usable web App to AmadeusGPT using GPT-3.5 with the 3 demo datasets used in the submitted manuscript (EPM, MABe, and MausHaus videos) and asked neuroscience behaviorists to test it – namely, we added example prompts they could run and a chat-interface to ask their own questions. Here, we now collected extra prompts not generated by the authors for testing and we will include this in the revision. In brief, we sampled 30 naive user prompts at random out of the 362 submitted via our App (prompts having caused a programming error are marked with a **X**), and found a roughly 18% error rate with GPT3.5 (see more below for GPT4, which was only 10%):

- **X** Get angles and distances between all body parts. Plot a UMAP graph using the resulting data.
- Give me the duration of time the animal engages with the object I picked and the events where the animal overlaps with the object I picked.
- **X** Perform hierarchical clustering and plot the dots with different colors based on their clusters.
- Define <|sap|> as a behavior where the animal's body length elongates. Do you see sap in the epm?
- Give me the tracking of all 3 animal's noses with each nose being shown in a different color.
- Define $< |peeking| >$ as a behavior where the animal's head_midpoint is not in ROI5 or ROI4 and the mid_back point is in ROI5 or ROI4.
- **X** Plot the distance between animals over time.
- **X** What is the speed of the changes from the left to the right arm?
- Plot regions of interest (ROIs).
- **X** Define $< |relative\_head_angle| >$ as the angle between the mouse_center and the head_midpoint. Plot the variation of $< |relative\_head\_angle| >$ over time.
- How many animals are there in this data?
- Give me the keypoints.
- Give me the total distance traveled by each animal.
- Define $< |watch| >$ as a social behavior where distance between animals is less than 260 and larger than 50 and head angle between animals is less than 15. The smooth_window_size is 15.
- **X** Plot animal center of mass x-coordinate, velocity, acceleration, and head direction.
- Define $< |freezing| >$ as a behavior where the animal is in the same position for longer than 10 seconds.
- Plot a figure comparing the time the animal spends in the open arm (ROI0) and the time the animal spends in the closed arm (outside ROI0).
- **X** Plot a bar graph with the first bar representing the total time the animal spends in ROI0 (open arm) and the second bar representing the total time the animal spends outside of ROI0 (closed arm).

- Calculate the number of successful spontaneous alternations using a sliding window calculation. A successful spontaneous alternation is defined as when, in the last 5 arm entries, all four arms (ROI0, ROI1, ROI2, and ROI3) are visited at least once.
- **X** Plot the euclidean distance between the nose points of animal0 and animal2 over time.
- Why is the angle limited to 180 degrees instead of 360?
- Show which what? Please provide more information or specify your request.
- **X** Define $< |head\_direction| >$ as the orthogonal angle to the line between left_ear and right_ear.
- Give me the amount of time the animal spends in ROI0 and the events where the animal overlaps ROI0.
- Calculate head direction using the ear points and plot it in radians.
- **X** Plot each grooming bout.
- Define $< |head\_scans| >$ as a behavior where the animal's head-direction shifts by more than 15 degrees to one side and then the other within 5 seconds. Give events where head_scans happen and bouts for head_scans.
- "Find events where there are errors in tracking due to outliers in mean distance between all points, animal acceleration, and velocity. Remove the labeled point causing the error and fill in missing values by interpolation. Plot the x and y coordinates of the animal's center of mass before and after the tracking correction.
- Give me the frequency of rearing events for the animal.
- Print overlap_roi0_events and its items.

We analyzed the logs from 362 queries, 242 of which were from unique prompts (i.e., from manually typed ones rather than executing the demos). Out of the 362 queries, 329 were automatically rephrased. We found AmadeusGPT had reported 129 "errors": 38 were caused by unsupported features or undefined variables, and were thus explained to the user, while 64 originated from programming errors. Therefore, from 362 queries there were an 18% programming error rate, and 11% unsupported feature requests rate.

Then, we took 10 of the failed runs from external users and re-ran them with GPT4. Now, of the 10 that previously failed, only 3 failed, and 4 correctly output clarifying queries to the user*; This means that if GPT4 was used, given the randomly sampled prompts, the error rate would be only 10

- **X** Get angles and distances between all body parts. Plot a UMAP graph using the resulting data.
- Perform hierarchical clustering and plot the dots with different colors based on their clusters.
- Plot the distance between animals over time.
- What is the speed of the changes from the left to the right arm?
- Define $< |relative\_head\_angle| >$ as the angle between the mouse_center and the head_midpoint. Plot the variation of $< |relative\_head\_angle| >$ over time.
- **X** Plot animal center of mass x-coordinate, velocity, acceleration, and head direction.
- Plot a bar graph with the first bar representing the total time the animal spends in ROI0 (open arm) and the second bar representing the total time the animal spends outside of ROI0 (closed arm).
- Plot the euclidean distance between the nose points of animal0 and animal2 over time.
- **X** Define $< |head\_direction| >$ as the orthogonal angle to the line between left_ear and right_ear.
- Plot each grooming bout.

*For (2), (4), (5), (10), the output is a message indicating either that the feature is not implemented or that the query does not provide all information, and explains what would be needed either in terms of development or clarifications from the user.

**Example from Explainer**

Here is an example query on the EPM dataset and relevant outputs for the explainer module:

```
Before rephrasing: How much time does the animal spend in the closed arm which is ROI0?

---- function code -----
def task_program():
    behavior_analysis = AnimalBehaviorAnalysis()
    overlap_closed_arm_events = behavior_analysis.animals_object_events('ROI0',
    ↪  'overlap', bodyparts=['all'])
    return overlap_closed_arm_events

generated explanation
1) The code `behavior_analysis = AnimalBehaviorAnalysis()` creates an instance of the
↪  AnimalBehaviorAnalysis class, which is a tool for analyzing animal behavior.

The next line `overlap_closed_arm_events =
↪  behavior_analysis.animals_object_events('ROI0', 'overlap', bodyparts=['all'])` uses
↪  this tool to find events where the animal overlaps with the closed arm region of
↪  interest (ROI0). It does this by calling the `animals_object_events` method of the
↪  AnimalBehaviorAnalysis class, passing in the parameters 'ROI0', 'overlap', and
↪  ['all']. This method uses the API documentation to access the necessary data and
↪  perform the analysis, returning the events where the animal overlaps the closed arm.

2) The meta answer is: For animal0, the total queried events last 7884 frames (total
↪  video length is 15076 frames).

This means that out of the entire video, there were 7884 frames where the animal
↪  overlapped with the closed arm ROI0. The total video length is 15076 frames, so the
↪  queried events make up a significant portion of the video.

Analyzing this in terms of animal behavior, it suggests that the animal is spending a
↪  considerable amount of time interacting with or being near the closed arm.
```

**Example from Self-correction**

Here is an example query on the MausHaus dataset and relevant outputs for the self-correction module:

```
Query: Plot trajectory of the center of the mouse body

def task_program():
    behavior_analysis = AnimalBehaviorAnalysis()
    traj_plot_info = behavior_analysis.plot_trajectory(["center"])
    return traj_plot_info

ValueError: center not defined in the data. Supported bodyparts are ['nose', 'left_ear',
↪  'right_ear', 'left_ear_tip', 'right_ear_tip', 'left_eye', 'right_eye', 'neck',
↪  'mid_back', 'mouse_center', 'mid_backend', 'mid_backend2', 'mid_backend3',
↪  'tail_base', 'tail1', 'tail2', 'tail3', 'tail4', 'tail5', 'left_shoulder',
↪  'left_midside', 'left_hip', 'right_shoulder', 'right_midside', 'right_hip',
↪  'tail_end', 'head_midpoint']

Diagnosis: The error message indicates that the variable "center" is not defined in the
↪  data. The supported body parts for plotting the trajectory are listed in the error
↪  message.

To fix this issue, you need to make sure that the data contains the "center" body part.
↪  You can check if the data has the required body part before plotting the trajectory.

revised function code:
```

```python
def task_program():
    behavior_analysis = AnimalBehaviorAnalysis()
    traj_plot_info = behavior_analysis.plot_trajectory(["mouse_center"])
    return traj_plot_info
```

## Horse Gait Analysis

### Horse gait analysis demo

👤: Query:
"Could you tell me what the duty factor of the Offfrontfoot is and explain its value to me?"

🪰 Chat Output:

0.59
The duty factor of the Offfrontfoot is greater than 0.5, indicating that the animal spends more time in the stance phase with its front foot than in the swing phase.

👤: Query:
"ok! Could you also calculate the duty factors using the keypoint 'Offhindfoot', and compare both?"

🪰 Chat Output:

```python
def task_program():
    behavior_analysis = AnimalBehaviorAnalysis()
    offfrontfoot_analysis = behavior_analysis.run_gait_analysis(limb_keypoint_names=["Offfrontfoot"])
    offhindfoot_analysis = behavior_analysis.run_gait_analysis(limb_keypoint_names=["Offhindfoot"])
    offfrontfoot_duty_factors = offfrontfoot_analysis["duty_factors"]
    offhindfoot_duty_factors = offhindfoot_analysis["duty_factors"]
    comparison = np.mean(offfrontfoot_duration) - np.mean(offhindfoot_duration)
    return comparison
```

👤: Query:
"Could you explain what that means?"

🪰 Chat Output:

The duty factor is a measure of the amount of time a limb spends in contact with the ground during a gait cycle. It is calculated by dividing the stance duration (time when the limb is in contact with the ground) by the stride duration (time for a complete gait cycle).

By comparing the duty factors of the Offfrontfoot and Offhindfoot, we can determine if there is any difference in the amount of time these limbs spend in contact with the ground. A positive value indicates that the Offfrontfoot has a longer duty factor, meaning it spends more time in contact with the ground compared to the Offhindfoot. A negative value indicates the opposite, with the Offhindfoot having a longer duty factor."„

example of the Web App interface

👤: Query:
"Could you rerun the previous analysis but plot it this time? The limb is defined with the following keypoints: Offfrontfoot; Offfrontfetlock; Offknee; Elbow and Shoulder."

```python
def task_program():
    behavior_analysis = AnimalBehaviorAnalysis()
    analysis = behavior_analysis.run_gait_analysis(limb_keypoint_names=["Offfrontfoot"])
    fix, ax = behavior_analysis.plot_gait_analysis_results(
        analysis,
        limb_keypoint_names=["Offfrontfoot, Offfrontfetlock, Offknee, Elbow, Shoulder"])
    return fig, ax
```

Figure 8: **Horse Gait Analysis.** Examples of user queries and AmadeusGPT outputs, including example of the Web App interface. Data adapted from [36].

