# OpenReview forum: "AmadeusGPT: a natural language interface for interactive animal behavioral analysis"
_NeurIPS.cc/2023/Conference — NeurIPS 2023 poster_

### Official Review · Reviewer_b8BG · 2023-06-16

**Soundness:** 4 excellent
**Presentation:** 4 excellent
**Contribution:** 4 excellent
**Rating:** 8
**Confidence:** 4

**Summary:**

This paper presented Amadeus, a natural language interface for interactive animal behavior analysis. To accommodate modern LLM (GPT3.5) for behavior analysis, the authors proposed to use an API document to constrain GPT3.5’s knowledge space. Furthermore, the authors proposed Dual Memory Mechanism to read and write behaviors (symbols) to enable correct long-term memory retrieval and overcame the limit of 4096 tokens. The authors demonstrated results on three standard mouse behavior datasets.

**Strengths:**

(1) The motivation is quite novel. The whole system could provide an unprecedented experience of animal behavior analysis through natural language guidance only. This paper pioneers a practical way to integrate LLM to task programming.

(2) The paper is well written.

(3) The quantitative result on MABE dataset demonstrates the effectiveness of the system.


**Weaknesses:**

(1) Although the paper has claimed “animal behavioral analysis”, the datasets used are mouse only. I wonder whether it is difficult to transfer the experience of this paper to other animals, e.g. monkeys, zebrafish, etc. The authors should discuss its applicability to other animals.

(2) I want to know how many behaviors have been tested on this model. It would be better to list them in the supplementary files.


**Questions:**

(1) Can it extrapolate to behavioral codes that are never seen, and how are the atomic APIs distributed? I mean, the behaviors that could be handled seem to closely related to the capability of atomic API. If the atomic API could not measure animal overlap, would the system work with behavior like "mount" which is closely related to pixel/keypoint/area overlap?

(2) What if define a symbol twice with slightly different or totally different descriptions?

(3) Mistake:
In Figure.3, “Cmpute the pose from the video” -> “Compute …”


**Limitations:**

The authors have addressed limitations.

---

> ### Author Rebuttal · Authors · 2023-08-09
>
> Firstly we greatly thank you for your rating and recognition of the novel approach and performance of AmadeusGPT. Below we address your questions the best we can, and also agree it will be very exciting to see this deployed on more behaviors.
>
> To address your noted **weaknesses**:
>
> - **(1)** There is nothing limiting the system to mice (i.e. core modules are SAM (generalist), animal pose with SuperAnimal (Ye et al. 2023 released weights that work on over 45 species therefore we used them), and CEBRA with also is a generalist new dimensionality reduction algorithm just published in Nature (Schneider et al 2023, code publicly released March 2023). We do agree perhaps the choice of only mice tasks was limiting, but these are the most common benchmarks. However, we have now tested another species for you. In particular, we provided a new demo with a horse video. We added local maxima detection and some atomic gait analysis and visualization functionalities, and demonstrated that with an addition of <100 lines of code AmadeusGPT can be asked, for example, to evaluate the symmetry of a horse’s stride (see PDF in overall rebuttal for the visual demonstration).
>
> - **(2)** The Sturman et al, benchmark data (EPM), the 9* MABe challenge tasks we show, and the MausHaus video that allows us to demo the SAM + animal pose estimation, and integrations with CEBRA. Of course in the future we aim to open this code up to users and have a web portal – there is nothing inherent in SAM, pose models, or CEBRA that limit this to mice, and we focused on spatio-temporal reasoning as from flies to macaques, interactions with conspecifics and objects are present (and see above).
>
>  *-*We added 3 more tasks in response to another reviewer.*
>
>
> To respond to **questions**:
>
> - For **question (1)**, we agree the way we handle behaviors seems to be closely related to the capability of atomic API. Though it should only augment the underlying LLMs. In the early development stage, we found that many behavior tasks cannot be solved by naive GPT-3.5, possibly due to the lack of high quality training data for behavior analysis. Therefore, we tried to and will cover more atomic APIs by having behaviorists try our software.
>
> - Also, as we briefly mentioned in the text, this is a practical design choice for extensively using those atomic API or combinations of them. This is due to the limited code generation ability of GPT-3.5 and its context window. Adding constraints using atomic APIs, we are able to (1)  reduce code hallucinations, (2) reduce token consumption of the LLMs,  (3) encapsulate lots of pre / post processing that are needed for handling noisy real-world data. While more powerful LLMs such as GPT-4  (context window much bigger) becomes more available and cheaper, we will gradually relax the constraints from atomic APIs in the future. We also note behavior analysis is also subject to domain knowledge and the available LLMs might not be sufficiently trained to know these domains. Therefore, some constraints of atomic APIs will still remain even in the future. To your more specific question about “mounting” and “overlapping”, we do not see any problem for AmadeusGPT to handle overlapping and mounting, as they can be easily captured by the  distance between animals. Though 3D keypoints and other kinematics should be needed to well distinguish two behaviors. However, imagine it could not handle overlap, then it probably wouldn’t work for mounting as mounting and overlapping will involve the generalization or specialization of the same set of atomic APIs.
>
>
> - For **question (2)**, we have a mixture strategy for long term memory, you are welcome to check section 3.3 and section 3.5 for more details. In short, for dynamic module loading, we rely on embedding of the query for storing and retrieving. In that case, a slightly different description might cause false retrieval. But since you are asking about  our symbolic pointer approach for long term memory,  defining a symbol twice is equivalent to overwriting an entry in Python’s dictionary. And whether the descriptions are similar or very different should not impact the retrieval as we used regular expressions to look for the symbol in <> and <||>. This way, the description is guaranteed to be retrieved correctly as long as the symbolic name is given correctly.
>
> - For **question (3)** We will fix the typo in the figure, apologies! Mistake: In Figure.3, “Cmpute the pose from the video” -> “Compute …”

---

> > ### Comment · Reviewer_b8BG · 2023-08-11
> > **reply**
> >
> > I have read the rebuttal. I would like to maintain my initial rating. Good job.

---

> > > ### Author Response · Authors · 2023-08-15
> > >
> > > Thanks very much!

---

### Official Review · Reviewer_f7B3 · 2023-06-19

**Soundness:** 3 good
**Presentation:** 4 excellent
**Contribution:** 3 good
**Rating:** 6
**Confidence:** 3

**Summary:**

This paper proposes AMADEUS, a GPT3.5 powered system to perform animal behavior data analytics given a natural language user-given query and a video depicting animal behavior. The model works by using GPT-3.5 to generate python code which makes calls to a instance segmentation and animal pose model, as well as hard-coded modules to compute animal-object relationships and operates with their outputs to return results according to a given query. The proposed model is tested with 3 popular animal behavior analysis benchmarks, showcasing different domains, queries and tasks, showing behavior consistent with human annotators and surpassing existing baselines.




**Strengths:**

- Related work: The paper is open about closely related work, and assigns credit for the contributions that are used in the AMADEUS system.

- Method:
	- I appreciate the discussion about the design of the API. It would be valuable to understand what would be the effect of not having made the specific choices mentioned in lines 150-159.
	- I appreciate the Dual Memory Mechanims proposed (includeing the illustrative example of limitations of short context windows). I would like to understand the main differences between that appraoch and the one used in Generative Agents, and a discussion on why their execution flow is more expensive. The memory mechanism also has valuable usability features, such as being able to store and retreive states.
- Relying on LLM allows to provide complex data analysis queries that were not possible in previous methods, as well as follow up questions to a given data analysis.
- The proposed method, besides flexible is high performing in existing benchmarks, consisting on varied types of queries, expected outputs and data domains. The method perorms above the state of the art, both qualitatively adn quantitatively, surpassing all submissions in the MABE 2022 behavior challenge.



**Weaknesses:**

Novelty:
- The work is very similar to existing works using LLMs for computer vision tasks (ViperGPT, VisProg), with the main difference being that they are applied to the animal behavior domain. This is however reported in the related work section, and the work includes other components to handle long duration data.

Scope:
- This looks more like a systems paper, with many of the contributions already existing in previous works, and the main contribution being in combining these existing contributions for a new domain. I think that the paper does have value, but I am wondering if it is the right scope for NeurIPS.

Method:
- The performance on the method seems highly dependent on the data analysis modules, which are handwritten. How general are these modules to different kinds of animal behaviors?

- I am suspicious of the robustness and stress testing section. The full system relies on GPT3.5, and that same model is used to stress-test the system. I would like to see how good is the performance (with/without rephraser) from independent researcher-given queries.

**Questions:**

NA

**Limitations:**

Limitations are mostly addressed. The biggest missing limitation is how general is this approach to other animal behavior analysis, given the strong dependence on the hand-designed modules.

---

> ### Author Rebuttal · Authors · 2023-08-09
>
> We thank the reviewer for pointing out the high performance, flexibility, and new additions that our work brings that related work have not addressed yet (dual-memory), and also we thank you for praising our related works- its a quickly moving and exciting  time for LLMs+computer vision! We have worked to address your noted weaknesses and address your questions below.
>
> Also re: relation to Generative Agents by Park et al 2023  came out on arXiv only 30 days before our work, so while it has some similarities we did not compare it head-to-head. The largest difference we see is it builds synthetic agents with code, while we work to analyze real-world video data directly (this is not a critique of that very cool work). Regarding the expense of the execution flow, since our symbolic pointer for long term memory is implemented using Python dictionary (i.e., hash table), the computational complexity for the lookup operation and insertion are O(1). In contrast, generative agents require LLM to calculate the embedding for every query and the embedding will later be used to retrieve relevant content in a vector database where the lookup operation likely takes O(Nd) where N is number of vector in the database and d is dimension of the vectors.
>
> To address your noted **weaknesses**:
>
> - (1, 2) Novelty & Scope: We agree there are great synergies with ViperGPT and VisProg, but there are some differences and we do cite the work. The technical difference that you also highlight as a strength is our novel dual-memory system, which we do think is broader than animal behavior. Yet, given neurips is at the intersection of ML and neuroscience, we felt this was also the proper venue for our work, we hope you agree!
>
> - (3) Method; we don’t fully constrain the LLMs based only our API, and we wrote the API to be quite “general” for laboratory animal behavior – of course, that is still a specialist approach – but we do think it shows a first applied real-world use of LLMs and SOTA modules in computer vision and machine learning that are linked by our system. Directly, it can be used on other animals, and also to address another reviewer we show this on another species, and note that nothing is hard-coded about mice, the hard-coded nature is about spatio-temporal reasoning, which spans animals. We hope that clarifies the aim and approach.
>
> - (4) We developed a web-App and invited external users to test the system with GPT3.5 models, thereby its OOD and stress tested.
> We analyzed the logs from 362 queries, 242 of which were from unique prompts (i.e., from manually typed ones rather than executing the demos). Out of the 362 queries, 329 were automatically rephrased.
>
> We found AmadeusGPT had reported 129 “errors”: 38 were caused by unsupported features or undefined variables, and were thus explained to the user, while 64 originated from programming errors. Therefore, from 362 queries there were an 18% programming error rate, and 11% unsupported feature requests rate.
>
> Then, we took 10 of the failed runs from external users and re-ran them with GPT4. Now, of the 10 that previously failed, only 3 failed, and 4 correctly output clarifying queries to the user*:
>
> 1: ['❌ Get angles and distances between all body parts. Plot a UMAP graph using the resulting data.',
>
> 2:  '✅ Perform hierarchical clustering and plot the dots with different colors based on their clusters.',
>
>  3: '✅ Plot the distance between animals over time.',
>
> 4:  '✅ What is the speed of the changes from the left to the right arm?',
>
> 5:  '✅ Define <|relative_head_angle|> as the angle between the mouse_center and the head_midpoint. Plot the variation of <|relative_head_angle|> over time.',
>
> 6:  '❌ Plot animal center of mass x-coordinate, velocity, acceleration, and head direction.',
>
> 7;  '✅ Plot a bar graph with the first bar representing the total time the animal spends in ROI0 (open arm) and the second bar representing the total time the animal spends outside of ROI0 (closed arm).',
>
> 8:  '✅ Plot the euclidean distance between the nose points of animal0 and animal2 over time.',
>
> 9;  '❌ Define <|head_direction|> as the orthogonal angle to the line between left_ear and right_ear.',
>
> 10:  '✅ Plot each grooming bout.',
>
> *For (2), (4), (5), (10), the output is a message indicating either that the feature is not implemented or that the query does not provide all information, and explains what would be needed either in terms of development or clarifications from the user.
>
> Thank you for this suggestion to stress-test further, and hope this overall response enables you to support our work!

---

> > ### Author Response · Authors · 2023-08-19
> >
> > Dear Reviewer f7B3,
> >
> > Are there any clarifications we can provide for you regarding our rebuttal? Thank you for your time and efforts in this busy period.

---

> > > ### Author Response · Authors · 2023-08-21
> > >
> > > Apologies in advance to bother you, but given today is the last period of rebuttal clarifications, we want to be sure you saw our rebuttal. Thank you!

---

### Official Review · Reviewer_eSvp · 2023-06-29

**Soundness:** 3 good
**Presentation:** 3 good
**Contribution:** 3 good
**Rating:** 6
**Confidence:** 3

**Summary:**

This paper proposes Amadeus, a novel natural language-based interface that leverages large language models like ChatGPT and vision-language models like SAM for animal behavior tracking and analysis. Amadeus leverages LLMs to generate code outputs, which can be executed to retrieve visual information and memory information and return requested responses from user queries. In particular, Amadeus proposes a novel dual-memory mechanism that combines short-term and long-term memory banks to effectively analyze extended contexts such as long videos. Experiments demonstrate that Amadeus achieve state-of-the-art performance on the MABE Behavior Challenge 2022 benchmark.

**Strengths:**

- The paper is generally well-written, and the figures are generally informative about the details of each module of Amadeus.

- Leveraging recent advancements on LLMs and VLMs to analyze animal behaviors is a novel idea and holds significant potential for scaling up to more complex animal behaviors and a larger number of animals in the future. One of the strengths of this approach is its accessibility and training-free nature, which makes it adaptable to novel scenarios with relative ease.

**Weaknesses:**

In Fig. 4, the implementation of long-term memory storing and retrieval processes is a bit unclear, as the generated code does not explicitly store or retrieve information into / from the long-term memory bank.

While the proposed approach achieves the state-of-the-art results on the MABE Behavior Challenge 2022 benchmark, it would be beneficial to include discussions on the failure cases, and analyze their source of failures (e.g., due to mistaken VLM perceptions, or due to mistaken code generations, or due to compounding errors caused by the wrong values pushed to the memory banks, etc.)

**Questions:**

See "weaknesses"

**Limitations:**

Limitations are mostly sufficiently addressed. Another limitation to add is that the current approach is bottlenecked by the capabilities of LLMs and VLMs (e.g., SAMs) being used. Perception errors can occur due to the limited capabilities of current VLMs, and the ability to produce correct responses given more complex queries and more demanding tasks is bounded by the capabilities of current LLMs.

---

> ### Author Rebuttal · Authors · 2023-08-09
>
> We thank the reviewer for noting our novel idea and future potential to scale more broadly, we agree and are excited about this prospect. We also appreciate your noted weaknesses and do our best to address them.
>
> To address your noted **weaknesses**:
>
> - We will clarify Figure 4 by noting the generated code is examples without the memory (i.e., it makes mistakes) vs. with the memory generated code, and additionally add a diagram that makes the flow clear (add <|keyword|>, this goes to memory so tokens can be used otherwise, and recall with keyword).  Conceptually, the storing and retrieving process are implemented as a default system behavior that is being executed silently at every iteration of the question-answering and we described the implementation details in section 3.3 and section 3.5. To clarify further, the reason we implemented storing and retrieving as a system behavior instead of API calls in the generated code is that we did not find them necessary to be included into APIs because they do not need to adapt to user inputs and we tried our best to save token consumptions.
>
> - We will also show mistakes in the Appendix, as of course it does not reach 100% performance on the task, and this can come in two flavors: (1) Amadeus outputs unusable code, or (2)  the code cannot capture all the behaviors in the video. Based on external testing, the rate of code failure was 18% and 11% unsupported feature requests with GPT3.5, and only 10% errors with GPT4 (note we used GPT3/5 for MABe results). For MABe results, we ruled out failure cases caused by errors of code generation because we have self-correction mechanism that can fix obvious code generation errors for social behavior APIs used in MABe and we cache the correct code when iterating the code through 3000+ video data (this is also, of course, to save the budget of running tens of thousands of ChatGPT API calls).
>
> We also agree in general with your noted limitations of LLMs, but hope these additions and clarifications will help you increase your rating of our work!

---

> > ### Comment · Reviewer_eSvp · 2023-08-10
> > **Reply**
> >
> > Thanks authors for the rebuttal! My concerns have been sufficiently addressed and I keep my original rating.

---

> > > ### Author Response · Authors · 2023-08-15
> > >
> > > Thanks very much!

---

### Official Review · Reviewer_ZigY · 2023-06-30

**Soundness:** 3 good
**Presentation:** 4 excellent
**Contribution:** 2 fair
**Rating:** 5
**Confidence:** 3

**Summary:**

This paper presents a novel interface and deep-learning system that enables interactive animal behavioural analysis using only natural language, on tasks that previously required extensive coding expertise and technical machine learning knowledge. The proposed framework integrates various modules based on LLMs and existing vision networks dedicated to various parts of the task, such as understanding the data processing API and rephrasing user prompts. This work also contributes a novel long-short-term memory technique in the context of LLMs. In combination, the contributed interface and systems achieves state-of-the-art performance on established tasks in the domain of animal behavioural analysis, and has the potential to support domain experts in using neural techniques and software frameworks that are currently difficult for them to use.

**Strengths:**

- Enables the novel user experience of analyzing animal behaviour using only natural language, which could be an important affordance for domain experts

- Well-designed, workable end-to-end system that reasonably combines multiple deep-learning modules, all with dedicated purposes in various aspects of the task.

- Effective use of SoTA techniques in deep learning to achieve high performance on established domain-specific tasks

- Introduced a novel long-short-term memory module that is compatible with LLMs and shown effective usage of such module in the contributed framework

**Weaknesses:**

- **Unclear general applicability of the long-short-term memory module and the framework in general:** From a machine learning perspective, the most significant technical contribution would be the long-short-term memory module in the context of LLM that the author(s) developed. The current paper discusses this module primary in the context of animal behavioural analysis. I would like to see some discussion of the general applicability of the memory module to common domains utilising LLMs, and also the general applicability of the entire framework to other domains, such that researchers working on those domains can benefit from the technical contributions of this paper.

- **Unclear generalizability of the work due to author-crafted prompts:** This is a limitation the author(s) have acknowledged, such that the experiments performed by the author(s) all followed prompts crafted by the author(s). I also acknowledge that the author(s) have attempted to alleviate this issue by introducing OOD base questions and using similar task descriptions as the original definition in the MABE challenge. However, I am curious about the opinion of the end-user (i.e., animal behaviour analysts/specialists) on the ability of the system in handling their prompts. It would be great if the author(s) can include some study (could be formative/informal) and/or discussion regarding the ecological validity of the system. It would be great if the author(s) can report lower-level success metrics, such as the rate of having syntax/compilation errors of the analysis code generated by Amadeus.

- **Limited technical novelty and contribution for Machine Learning**: Related to the first point, the primary technical contribution of the paper in terms of modelling techniques is the long-short-term memory module, which is limited. I believe more extensive discussion and/or experiments on this module could be helpful for researchers in the ML community to build upon the findings of this paper.

**Questions:**

1) What are the general applicability of the long-short-term memory module and the entire framework to domains other than animal behaviour analysis?

2) Did any animal behaviour experts tried using the system / reviewed the prompts in the experiments of this paper? If so, what are their opinions on the ecological validity of them?

3) What is the more low-level success metrics of the code generated by the system, such as the compilation error rate?

**Limitations:**

The limitations of this work were adequately discussed by the author(s).

---

> ### Author Rebuttal · Authors · 2023-08-09
>
> Firstly we would like to thank the reviewer for pointing out the novelty of our overall system and the novel dual-memory system. We’d like to add that we aimed to be generous in related works, we are one of the first systems to show LLM-generated executable code and error correction, and our Rephraser model we had not seen before (of course since early May more works have come out).  Thus, we want to push back a little that this paper is just a summation of many parts. We hope below (and response globally and to other reviewers) will convince you this is a paper for not only the machine learning crowd, but also for neuroscientists.
>
> To address **weaknesses**:
>
> - **(1)**  This is a fair point that we only discuss or test our system in the context of animal behavior. We can easily speculate that this is much broader, there is nothing inherent about it being related to behavior, but we wanted to be careful to not over claim this in the manuscript.  To speculate, many applications that demand the system to retain distant context for tasks like answering questions or ongoing learning. Thus, we believe the general capability of this framework is also a good contribution to people working in other domains.  Our modular design and philosophy of the atomic APIs vs. integration APIs can provide interesting references to people who want to build a flexible, extendable analysis framework using LLMs and machine learning models and open source libraries.
>
> - **(2)**  To probe the system we now have released a fully usable front end user interface to Amadeus with the 3 demo datasets used in the submitted manuscript (EPM, MABe, and MausHaus videos) and asked neuroscience behaviorists to test it - -namely, we added example prompts they could run and a chat-interface to ask their own questions. Over 100 signed up in 48 hours, and we admitted 30 as alpha testers who stated they worked with laboratory animal behaviors.
>
> We analyzed the logs from 362 queries, 242 of which were from unique prompts (i.e., from user typed rather than executing the demos). Out of the 362 queries, 329 were automatically rephrased.  We found Amadeus had reported 129 “errors”: 38 were caused by unsupported features or undefined variables, and were thus explained to the user, while 64 originated from programming errors. Therefore, from 362 queries there were 18% programming errors, and 11% unsupported feature requests.
>
> The users also had the option to give a “like”  or “dislike” about if the output matched their expectations, and out of 51 responses 6 dislikes reported to the fully-executed outputs, 18 dislikes to code errors, which of course is expected, and 33 likes. Thus, while of course there is room for improvement, we hope this generally addresses your excellent point on generalizability to real-world users.
>
> - **(3)** We do appreciate that our main technical contributions relate to the novel dual-memory system, but we also think the whole integrated system of multi-LLM models working together to write, refine, and explain results after the python interpreter is sufficiently novel, especially at the intersection of neuroscience and machine learning. We certainly will add to the discussion/paper about other domains of interest for this system-level approach too. We will also add more about our memory module:
>
> Implementing short-term memory via a deque for LLMs is straightforward. However, our focus is on long-term memory. Two main approaches exist: 1. Altering the transformer architecture, 2. Using external storage like vector databases seen in generative agents and recent works. Approach 1, altering architecture, is the natural way to extend LLMs' memory span. It requires further investigation to determine if the extended span affects language task performance, a topic we plan to explore. Note that this might incur higher costs compared to GPT-3.5 API calls. For Approach 2, seen in projects like Auto-GPT, a vector database serves as external storage for context embeddings. This method can be expensive due to embedding computations and a vector database lookup operation complexity of O(Nd), N being vectors and d their dimension.
>
> Our long term memory is using a mixed strategy: For long text such as source code, our dynamic module loading is similar to those works that store and look up embeddings using vector databases. The dynamic module loading fetches the most relevant integration module based on the query of the user, providing a flexible way to collect useful integration modules from the community (2). However, we also noted that for interactive behavior analysis, if the user just wants to retrieve some content such as the definition of the behavior,  our symbolic pointer-based long-term memory is also an effective solution. Firstly, using symbolic pointers,  there is no need for LLMs to calculate the query embedding every time, thus lowers financial and computational burdens. Additionally,  as definitions of many behaviors are highly similar in text and fetching context based on embeddings of description can be error prone.  Our symbolic pointer approach uses regular expressions to look for what is in the special tags such as <> and <||>. It guarantees that the fetched description is correct as long as users keep track of the used symbol names, as programmers need to remember the variable names. Secondly, since our symbolic pointer approach implements long-term memory using a hash table, it provides cheap insertion and lookup operation with complexity O(1), a sharp contrast to O(Nd) with a vector database.
>
> In summary, we are one of the first works that propose long-short-term memory for this kind of system and our mixture strategy for long-term memory is a good practical solution. We hope the above responses answered your **direct questions** (see points **3**, and **2** respectively). We thank you for your expertise, and hope our new analysis and clarifications enable you to find our work now acceptable, thank you!

---

> > ### Author Response · Authors · 2023-08-18
> >
> > Dear Reviewer ZigY,
> >
> > Are there any clarifications we can provide for you regarding our rebuttal? Thank you for your time and efforts in this busy period.

---

> > ### Comment · Reviewer_ZigY · 2023-08-18
> >
> > Thank you for the author(s) detailed rebuttal and the extensive work in testing involving real expert users, which addressed some of my concerns. As a result, I raised my score from 4 to 5.
> >
> > I suggest the author(s) to include these changes in their revised paper, if accepted.

---

> > > ### Author Response · Authors · 2023-08-18
> > >
> > > Thanks very much!

---

### Official Review · Reviewer_XjKy · 2023-07-06

**Soundness:** 2 fair
**Presentation:** 3 good
**Contribution:** 3 good
**Rating:** 5
**Confidence:** 4

**Summary:**

This submission introduces a system, Amadeus, for performing behavioral analysis on animal videos. This system combines three elements: an API with descriptive docstrings for performing common behavioral analysis tasks, a modified version of GPT3.5 with an enhanced context window, and a prompt tuning of this GPT3.5 to allow for construction of task programs for behavioral analysis using a natural language interface. What is claimed is that (1) the language model can extend beyond its nominal 4096 character token window (2) the system allows for behavioral analysis with no additional code written (3) the system is easily extensible. The authors’ provide schematics of the system architecture and the integration between various components. There are numerous examples for how written code is produced from natural language. There are a few evaluations provided (1) comparing human raters and the amadeus system in computing time spent in arms of an elevated plus maze (2) comparing the performance of the system in a select number of tasks from the MABE challenge and (3) comparing the ability of a trained rephraser model to domain adapt queries to the developer’s syntactical patterns. There is a fairly comprehensive inclusion of the API in the appendix as well as examples of the rephraser module.

Overall I found the contribution novel, of potential immediate utility to academic neuroscience labs performing behavioral analysis, and as an example for other groups hoping to automate analysis in other systems. I however found the evaluations and description of the methods somewhat lacking, and the generalizability and extensibility of the somewhat unclear. These would need to be improved for me to strongly endorse the paper.


**Strengths:**

•	This is the first LLM integration I have seen in the neuroscience and I think the approach is potentially interesting. In some ways I think it could be even more interesting for codebases that analyze on standardized experimental equipment (e.g. fluorescence imaging, neuropixels) because the set of tasks to perform is arguably more standardized than behavioral analysis, where the approaches, species, and questions can be fairly diverse.

•	Many researchers in the life sciences do not have a formal background in writing code or have to interact with a fairly specialized API to analyze data, and so assistance is useful here.

•	The system schematics and examples are clear.

•	The API documentation is helpful, and the integration of state of the art systems like SAM into an open-source (if this is to be distributed) codebase is helpful.


**Weaknesses:**

•	The manuscript is missing descriptions of the system architecture and training details. It is difficult to fully understand how the system was trained without this.

•	The evaluations are somewhat cursory. The results on the EPM are hard to evaluate because the ‘ground truth’ from the human raters is quite variable. The MABE evaluation is only presented across a subset of tasks.

•	The generalizability and extensibility of the system to new users and new behavioral tasks is unclear. Part of this is the rephraser example is somewhat limited, but it is also unclear how successful new users will be in writing functions with appropriate documentation and that documentation hints are not interdependent.


**Questions:**

•	Table 1 only lists a subset of MABE tasks, the model results should be reported across the full benchmark.

•	The lack of methods makes it difficult to know the details of the memory augmentation approach and the extent of training of the rephraser module (and how much one expects the later to generalize).

•	I found the memory augmentation result under-contextualized and other work in this space should be discussed. Augmenting LLM memory window is a subject under intense technological development and the contribution should be put into proper context, especially because it is a secondary result.

•	The EPM results are hard to evaluate because of inter-human variability. An experiment with clearer ground truth would help.

•	In general, hallucinations are a real problem for LLMs and I am not convinced by the given evaluations that they would not occur (e.g. for what fraction of a given call does the system produce the correct result). This is especially problematic for a system designed to be used by people who do not code and may not be able to debug problems.

•	The API is fairly expansive and includes example of most functions call and keyword arguments present in the given examples in the paper. This could mean that the system is limited in functionality to simply modifying parameters and numerical values given as input to the functions. Some statement about the expected generalization ability of the approach and the requirements for adding new functions to the docstring (e.g. does one have to provide examples of every new keyword input) is needed.

•	The claim that using the system would obviate the need to be able to code is strong. As the authors’ note, hallucinations are a problem for LLMs. Many in the field regard LLM solutions as an accelerated rough draft to be ‘fact-checked’ by a domain expert. Similar to the rephrasing analysis, it would be nice to have a statement about the reliability of the results, ideally across a pool of fairly naïve users.


**Limitations:**

Yes

---

> ### Author Rebuttal · Authors · 2023-08-09
>
> Thank you for noting the novelty and potential strength of our application to life sciences, and thanks for the suggestion to consider more experimentally constrained (i.e., imaging) settings, which we can try in the future. Here, we focused on classical behavioral tests, mostly for this reason, as many labs deploy these rather standardized setups worldwide (i.e., EPM).
>
> To address **weaknesses**:
>
> - **(1)** We do not fine-tune GPT3.5 (or GPT4) or the segmentation (SAM) or pose estimation models (SuperAnimal). Our system consists of three GPT3.5 instances with different system prompts, the pre-trained computer vision models, and our API implementation, we will clarify in revision. Re: system architecture, we hope Figure 2 gives a comprehensive design of the system, and more details about API design and system prompt design are provided in Appendix.
>
> - **(2)** EPM, and mouse behavior in general, is somewhat subjective, hence the absolute ground truth is a mixture of experts. While we fully appreciate this is not always satisfying, this is a real-world application and challenge Amadeus is likely to be faced with. MABe contains the experimentally-defined tasks and expert-annotated behavior tasks, of which 9 of 13 could likely be solved by pose estimation and spatiotemporal reasoning, and the other 4 are not feasible without a ML classifier, such as predicting the animal strain, etc. We now added the 3 other tasks that Amadeus could tackle, which we will add to the final version:
>
> |               | T4 approaches | T10 oral-genital-contact | T11 oral-oral-contact |
> |---------------|---------------|-------------------------|-----------------------|
> | PCA baseline  | 0.00          | 0.00                    | 0.00                  |
> | Top-entry 1   | 0.020         | 0.015                   | 0.013                 |
> | Top-entry 2   | **0.026**         | 0.029                   | 0.023                 |
> | Top-entry 3   | 0.022         | 0.015                   | 0.014                 |
> | BAMS          | 0.02          | 0.0165                  | 0.014                 |
> | Amadeus       | 0.014         | **0.05**                    | **0.05**                  |
>
> - (3) To expand on what the system can do we now have released a fully usable front end user interface App to Amadeus using GPT-3.5 with the three demo datasets used in the submitted manuscript (EPM, MABe, and MausHaus videos) and asked neuroscience behaviorists to test it – namely, we added example prompts they could run and a chat-interface to ask their own questions. We now collected extra prompts not generated by the authors for testing and we will include this in the revision. In brief, we sampled 30 naive user prompts at random out of the 362 submitted via our App (errors= ❌), and found a roughly 18% error rate with GPT3.5 (see more below for GPT4, which was only 10%); due to space limits, we show a few here:
>
>  '✅ Give me the duration of time the animal engages with the object I picked and the events where the animal overlaps with the object I picked.',
>
>  '❌ Perform hierarchical clustering and plot the dots with different colors based on their clusters.',
>
>  "✅ Define <|sap|> as a behavior where the animal's body length elongates. Do you see sap in the epm?",
>
>  "✅ Give me the tracking of all 3 animal's noses with each nose being shown in a different color.",
>
> From 362 queries 242 of which were from unique prompts (i.e., from user typed ones rather than executing the demos). 329 were automatically rephrased. We found Amadeus had reported 129 “errors”: 38 were caused by unsupported features or undefined variables, and were thus explained to the user, while 64 originated from programming errors. Therefore, from 362 queries there were an 18% programming error rate, and 11% unsupported feature requests rate.
>
>  **Questions**:
>
> - **1 & 2** are hopefully answered above. Note that the Rephraser LLM requires no training and the difference between the code generator and Rephraser is the system prompt, which is used for LLMs to do in-context learning; we will be sure to add the Rephraser prompts in a revised version.
>
> - **3**, we note that of course there is a lot of concurrent work right now, but we did our best to put in context and cite papers that we saw that were closest to ours at the time of submission (even if they did not impact our own ideas), but if you have concrete examples we missed, please let us know!
>
> - **4**, this is why we semi-constrain with the API, but of course we collectively need to understand issues with LLMs. New work that includes GPTs to write executable code are being tested rapidly, and it seems that GPT-4 give less hallucinations than GPT-3.5 (Cai et al. 2023 & Wang et al. 2023). Since our system can easily switch from GPT-3.5 to GPT-4, the hallucinations could be reduced with the improvement of the underlying LLMs used. However, our Rephraser truly does attempt to write code for 90% (ie 329 of 362 from outside users) of the time, i.e., tests if code is runnable and otherwise parses errors and tries again.  Therefore, we are happy to tone down any claims of “no coding ever,” but we do see this as a truly new way to interact with SOTA models (like SAM, etc), so we hope you agree this is still a useful demonstration for the field and neurips community.
>
> - **5**, we note this is still built on GPTs, so it’s not merely bound to our API. Our system prompts and API sets the ground rules to have it understand common behavioral analysis queries, but it doesn’t limit its capacity. As an additional test, we took 10 of the failed runs from external users and re-ran them with GPT4, which reduced errors to 10%, suggesting better LLMs also increase performance/our API is not too limiting.
>
> - **6**, see point 3 in weaknesses above, we have now asked expert mouse behaviorists to use the system.
>
> We hope this clarifies our novel contributions and improvements, and that you’d consider raising your rating. Thank you!

---

> > ### Author Response · Authors · 2023-08-19
> >
> > Dear Reviewer XjKy,
> >
> > Are there any clarifications we can provide for you regarding our rebuttal? Thank you for your time and efforts in this busy period.

---

> > > ### Comment · Reviewer_XjKy · 2023-08-19
> > > **further questions**
> > >
> > > Thanks for some of these responses. I appreciate the contribution and the time but still have a few open questions that could influence my score:
> > >
> > > > From 362 queries 242 of which were from unique prompts (i.e., from user typed ones rather than executing the demos). 329 were automatically rephrased. We found Amadeus had reported 129 “errors”: 38 were caused by unsupported features or undefined variables, and were thus explained to the user, while 64 originated from programming errors. Therefore, from 362 queries there were an 18% programming error rate, and 11% unsupported feature requests rate.
> > >
> > > Can you clarify the 242 number and what fraction of errors came from manually typed prompts? Understanding what fraction of the 129 errors came from the demo prompts vs. manually generated could substantially change the interpretation.
> > >
> > > > • The manuscript is missing descriptions of the system architecture and training details. It is difficult to fully understand how the system was trained without this.
> > >
> > > Are you planning on including details about the training procedure and more methodological descriptions of the networks used?
> > >
> > > > EPM, and mouse behavior in general, is somewhat subjective, hence the absolute ground truth is a mixture of experts
> > > this is a minor point. FIgure 4 a.1 is confusingly oriented such that the variance presented is across videos/trials which obscures any potential differences across humans and amadeus. For quantitative data its a very qualitative and anecdotal result of 'they are in a similar ballmark', but only using 5 trials and plotting the data to maximize trial to trial variance is a fairly poor way of comparing these.

---

> > > > ### Author Response · Authors · 2023-08-20
> > > >
> > > > >Can you clarify the 242 number and what fraction of errors came from manually typed prompts? Understanding what fraction of the 129 errors came from the demo prompts vs. manually generated could substantially change the interpretation.
> > > >
> > > > Yes, sorry for any confusion! None of the 242 were manually typed, these were all new, naive user inputs (from alpha testers). If we used GPT3.5 it was 18% overall (therefore 26% of the 242, 18% of the 362). Critically, we **also tested 10 failed runs from users (that used GPT3.5) and re-ran the prompts with GPT4, now only 3 failed.** Thus, this would be an 11% effective overall error rate. Here are examples of the 10 that previously failed with GPT3.5, where now only 3 fail:
> > > >
> > > > 1: ['❌ Get angles and distances between all body parts. Plot a UMAP graph using the resulting data.',
> > > >
> > > > 2: '✅ Perform hierarchical clustering and plot the dots with different colors based on their clusters.',
> > > >
> > > > 3: '✅ Plot the distance between animals over time.',
> > > >
> > > > 4: '✅ What is the speed of the changes from the left to the right arm?',
> > > >
> > > > 5: '✅ Define <|relative_head_angle|> as the angle between the mouse_center and the head_midpoint. Plot the variation of <|relative_head_angle|> over time.',
> > > >
> > > > 6: '❌ Plot animal center of mass x-coordinate, velocity, acceleration, and head direction.',
> > > >
> > > > 7; '✅ Plot a bar graph with the first bar representing the total time the animal spends in ROI0 (open arm) and the second bar representing the total time the animal spends outside of ROI0 (closed arm).',
> > > >
> > > > 8: '✅ Plot the euclidean distance between the nose points of animal0 and animal2 over time.',
> > > >
> > > > 9; '❌ Define <|head_direction|> as the orthogonal angle to the line between left_ear and right_ear.',
> > > >
> > > > 10: '✅ Plot each grooming bout.',
> > > >
> > > > **We will absolutely add this new analysis to manuscript 😃!**
> > > >
> > > > > • The manuscript is missing descriptions of the system architecture and training details. It is difficult to fully understand how the system was trained without this.
> > > > > Are you planning on including details about the training procedure and more methodological descriptions of the networks used?
> > > >
> > > > **Yes, of course we will add methodological details** about the networks and what we added in the first rebuttal, namely we did not train GPT3.5 or GPT4 ourselves, we rely on openAI API calls. Apologies if that was not clear in our first rebuttal: "(1) We do not fine-tune GPT3.5 (or GPT4) or the segmentation (SAM) or pose estimation models (SuperAnimal). Our system consists of three GPT3.5 instances with different system prompts, the pre-trained computer vision models, and our API implementation, we will clarify in revision. Re: system architecture, we hope Figure 2 gives a comprehensive design of the system, and more details about API design and system prompt design are provided in Appendix."
> > > >
> > > > Just to add, one feature we want to highlight is one can think of Amadeus like a front end to foundational LLMs and computer vision models that allow for customized APIs plus our new dual-memory system that requires no fine-tuning or vector database. Our rebuttal point 3 to reviewer ZigY might also be of interest on this latter point.
> > > >
> > > > > "EPM, and mouse behavior in general, is somewhat subjective, hence the absolute ground truth is a mixture of experts"
> > > > > -  this is a minor point. FIgure 4 a.1 is confusingly oriented such that the variance presented is across videos/trials which obscures any potential differences across humans and amadeus. For quantitative data its a very qualitative and anecdotal result of 'they are in a similar ballmark', but only using 5 trials and plotting the data to maximize trial to trial variance is a fairly poor way of comparing these.
> > > >
> > > > We apologize for any confusion here. What **we aimed to show is the same plot style as in the original benchmark paper** (see panel f here: https://www.nature.com/articles/s41386-020-0776-y/figures/4) Namely, those authors show that between 3 human raters they counted similar, but not the identical number of head-dips across 5 different long videos (not 5 trials) of mice. Each dot denotes a different mouse video). What we are trying to show, for example, is that all humans have the lowest count for the yellow mouse, as does Amadeus. All humans have the highest count for blue mouse, so does Amadeus. **We can happily modify the figure to connect the dots across humans and Amadeus to make that more obvious, thank you for the great suggestion.**
> > > >
> > > > We hope that clarifies your open questions and you'd consider supporting our paper. Thanks again for your time.

---

> > > > > ### Author Response · Authors · 2023-08-21
> > > > >
> > > > > Apologies in advance to bother you, but given today is the last period of rebuttal clarifications, we want to be sure you saw our last response.

---

### Author Rebuttal · Authors · 2023-08-09

Firstly we’d like to thank the reviewers, all who noted novel (or appreciable) advances with our use of LLM for behavior and our dual-memory system for LLMs:
- “Overall I found the contribution novel, of potential immediate utility to academic neuroscience labs performing behavioral analysis, and as an example for other groups hoping to automate analysis in other systems” - Reviewer XjKy
- “novel long-short-term memory module,” … “Enables the novel user experience of analyzing animal behavior using only natural language” - Reviewer ZigY
- “Amadeus proposes a novel dual-memory mechanism that combines short-term and long-term memory banks to effectively analyze extended contexts such as long videos” - Reviewer eSvp
- “The paper is open about closely related work, and assigns credit” … “I appreciate the Dual Memory Mechanisms proposed (including the illustrative example of limitations of short context windows)” … “The method performs above the state of the art, both qualitatively and quantitatively” - Reviewer f7B3
- “The motivation is quite novel. The whole system could provide an unprecedented experience of animal behavior analysis through natural language guidance only. This paper pioneers a practical way to integrate LLM to task programming.” - Reviewer b8BG

Secondly, we thank the reviewers for providing constructive feedback, which we address individually below.
In summary, the major points are:
- we provided a new analysis of the system from external users (here anonymous, but from behavioral experts)
- we added more behavioral examples beyond mice, i.e., we added horses, which is also attached as a new figure PDF
- we added three additional MABe challenge tasks

Overall, we are thankful for the reviewers support, and we hope these additions will improve your impression of our new system.

---

### Decision · Program_Chairs · 2023-09-21

**Decision:**

Accept (poster)

**Comment:**

The reviewers's sentiments are overall positive towards the paper. On the positive side, all the reviewers think the work is novel and addresses an interesting domain-specific problem (animal behavior analysis) with LLMs enhanced with long-and-short-term memory. Reviewer b8BG is the most enthusiastic, and commended that "The motivation is quite novel. The whole system could provide an unprecedented experience of animal behavior analysis through natural language guidance only. This paper pioneers a practical way to integrate LLM to task programming." Reviewer ZigY noted that (the work) "Enables the novel user experience of analyzing animal behaviour using only natural language, which could be an important affordance for domain experts". The reviewers also think the paper is well written. On the negative side, the reviewers have reservations with the generalizability, evaluation design and technical descriptions. Particularly, the work is only based on mouse data and its applicability to general animal behavior is not demonstrated. Nevertheless, the reviewers think the work makes a nice contribution to the problem.